# The origin and maintenance of supergenes contributing to ecological adaptation in Atlantic herring

Minal Jamsandekar [1,4], Mafalda S. Ferreira [2,4], Mats E. Pettersson [2], Edward D. Farrell [3], Brian W. Davis [1] & Leif Andersson [1,2,4] ✉

Chromosomal inversions are associated with local adaptation in many species. However, questions regarding how they are formed, maintained and impact various other evolutionary processes remain elusive. Here, using a large genomic dataset of long-read and short-read sequencing, we ask these questions in one of the most abundant vertebrates on Earth, the Atlantic herring. This species has four megabase-sized inversions associated with ecological adaptation that correlate with water temperature. The *S* and *N* inversion alleles at these four loci dominate in the southern and northern parts, respectively, of the species distribution in the North Atlantic Ocean. By determining breakpoint coordinates of the four inversions and the structural variations surrounding them, we hypothesize that these inversions are formed by ectopic recombination between duplicated sequences immediately outside of the inversions. We show that these are old inversions (>1 MY), albeit formed after the split between the Atlantic herring and its sister species, the Pacific herring. There is evidence for extensive gene flux between inversion alleles at all four loci. The large *Ne* of herring combined with the common occurrence of opposite homozygotes across the species distribution has allowed effective purifying selection to prevent the accumulation of genetic load and repeats within the inversions.

Chromosomal inversions suppress recombination in the heterozygous state, facilitating the maintenance of different combinations of alleles in tight linkage disequilibrium governing complex phenotypes, including the ones involved in local adaptation[1–4], reproductive strategies[5], life history traits[6], mimicry[7], and social behavior[8]. Sets of alleles within the inversion are inherited together as a single unit in Mendelian segregation and hence are also called supergenes[9]. Despite their evolutionary importance, the processes that lead to the origin, spread and maintenance of an inversion through time are often unclear because the evolution of inversion alleles is a dynamic process that changes over time and depends on the age, rate of gene flux, and effective population size ($N_e$) of both inverted and non-inverted haplotypes[10–13]. An inversion originates in a population as a single copy either by recombination between near-identical inverted duplication sequences, a process known as nonallelic homologous recombination (NHAR), or by a repair mechanism of a single-stranded break, known as Nonhomologous DNA End Joining (NEHJ) process[14–16]. Such processes can only be understood by characterizing the breakpoint region, which is notoriously difficult to study as it is often present in the highly polymorphic part of the genome surrounded by complex structural variations (SVs) and repeats[17,18]. Long-read sequencing makes it possible to uncover the complexity of breakpoints and shed light on the mechanisms forming inversions.

[1]Department of Veterinary Integrative Biosciences, Texas A&M University, College Station, USA. [2]Department of Medical Biochemistry and Microbiology, Uppsala University, Uppsala, Sweden. [3]Killybegs Fishermen's Organisation, Donegal, Ireland. [4]These authors contributed equally: Minal Jamsandekar, Mafalda S. Ferreira. ✉e-mail: leif.andersson@imbim.uu.se

Immediately after its formation, a single inversion copy is vulnerable to the effects of random genetic drift, whereby it either can be lost or increase in frequency[11]. If an inversion overlaps with co-adapted or beneficial allelic combinations, selection is likely to promote its maintenance and spread[11]. However, suppressed recombination in heterozygotes can result in impaired purifying selection and consequent accumulation of deleterious mutations in the inversion region, which theoretical and empirical data have demonstrated to ultimately result in the degradation of the inversion through the process of Müller's ratchet[6,7,19–23]. Interestingly, recent literature on vertebrate species supports the hypothesis that inversions can also evolve without pronounced accumulation of mutation load[24–28]. The accumulation of mutation load depends on several factors, such as age and frequency of an inversion haplotype, as well as the $N_e$ of a species, with more efficient purifying selection in large populations[10,12,13]. Furthermore, recombination may occur at low frequency in the heterozygotes, either through double crossover or gene conversion, facilitating purifying selection and purging of deleterious mutations[29,30]. Our study uncovers such a process and thus contributes to the understanding of the evolution of inversions.

In this study, we leverage the advancement in long-read sequencing technology with PacBio HiFi reads (average read length of 13.5 kb and accuracy above 99.8%) and use a large re-sequencing dataset to study four megabase-sized inversions on chromosomes 6, 12, 17, and 23 in the Atlantic herring (*Clupea harengus*) that are important for local adaptation[31]. The variant haplotypes at these loci are denoted Southern (*S*) and Northern (*N*), owing to their respective predominance in the southern and northern parts of the species distribution range in the northern Atlantic Ocean, possessing warmer and colder waters, respectively[31]. Atlantic herring is one of the most abundant vertebrates on Earth, with an $N_e$ over a million and a census population size ($N_c$) over a trillion, and has adapted to various ecological and environmental conditions such as variation in salinity, water temperature, light conditions, spawning seasons and food resources[31,32]. The effect of random genetic drift should thus be minute, with natural selection playing a dominant role in governing the evolution of genetic variation underlying ecological adaptation[31]. Thus, Atlantic herring is an excellent model to explore the evolutionary history of supergenes associated with inversions in natural populations.

Here, we used whole-genome PacBio HiFi data, combined with short-read Illumina data, from 12 Atlantic herring individuals, along with a previously generated high coverage re-sequencing dataset comprising 49 Atlantic herring and 30 Pacific herring (*Clupea pallasii*), the sister species[31,33] to shed light on (1) mechanisms of formation of inversion by finding breakpoint coordinates and structural variants (SVs) around the breakpoints, (2) the origin of inversions by describing its ancestral state and age using European sprat (*Sprattus sprattus*) as an outgroup species, (3) evolutionary history of inversions by phylogeny, (4) effects of suppressed recombination by analyzing patterns of variation, differentiation, linkage disequilibrium, mutation load, and gene flux (genetic exchange between inversion haplotypes).

## Results

### Samples and genome assemblies
The analysis of short-read as well as long-read data showed that the six Celtic Sea samples (CS2, CS4, CS5, CS7, CS8, and CS10) were homozygous for the Southern (*S*) allele for all four inversions (Fig. 1, Supplementary Fig 1). Among the six Baltic Sea samples (BS1–BS6), four were homozygous for all Northern (*N*) alleles, while two samples, BS2 and BS5, were heterozygous for the Chr23 and 17 inversions, respectively (Fig. 1, Supplementary Fig. 1). All samples were sequenced using PacBio HiFi with coverage ranging from 23x to 30x. We produced 24 haploid de novo genome assemblies from the 12 herring samples using hifiasm[34]. The two haploid assemblies per individual were denoted hap1 and hap2. All assemblies were of high quality, where genome size ranged from 743 to 792 Mb, the contiguity measured by N50 ranged from 452 to 737 kb and BUSCO scores were above 90%, indicating that PacBio assemblies contained more than 90% of conserved vertebrate genes (Table 1, Supplementary Table 1). Notably, hap1 assemblies had larger genome size and more contigs as compared to the hap2 assemblies. The unequal genome size for two haplotype assemblies suggests that a minor fraction of heterozygous sequences might not be accurately phased; while the positive correlation between genome size and number of contigs suggests that a small fraction of the genome is fragmented into multiple contigs.

We compared the quality of PacBio assemblies with that of the reference assembly of Atlantic herring (Ch_v2.0.2) and found that PacBio assemblies were of similar quality for genome size and BUSCO scores (Table 1). Although the total size of the reference assembly is 786 Mb, only 726 Mb is scaffolded in chromosomes ($n = 26$), and the remaining 61 Mb is present as unplaced scaffolds ($n = 1697$), i.e., fragments that could not be assigned to a chromosome. This material likely includes unresolved haplotypes, which is supported by the fact that all novel PacBio assemblies were above 726 Mb, indicating that these assemblies have higher portions of the heterozygous alleles resolved into separate contigs than its reference counterpart, which would be expected due to the improvement in accuracy provided by the HiFi technology.

### Characterization of inversion breakpoints on chromosomes 6, 12, 17, and 23
The PacBio HiFi read alignments (toward the reference assembly), in combination with the PacBio genome assemblies, were used to investigate the inversion breakpoints in detail. First, we used single-read alignments to accurately define inversion breakpoint coordinates. Here, the reads representing the alternate inversion are expected to show a particular pattern where the reads get split into two parts: one aligning outside the inversion and the other aligning inside the inversion at the opposite end in reverse orientation. Alignments of such PacBio HiFi reads are presented for each of the chromosomes harboring an inversion (see Fig. 2). Second, we used the PacBio genome assemblies to discern the sequences at the breakpoints leading to the formation of inversions. Figure 2 also illustrates the presence of inverted duplications flanking the breakpoints on Chr12 and 17. We manually inspected our data for such a pattern using IGV and Ribbon and determined the inversion breakpoints for all four inversions (six samples with alternate haplotypes for each inversion). We found similar, but not identical, breakpoint coordinates for each inversion across samples (Table 2), suggesting that these inversions have originated just once, stemming from a one-time break in the chromosome, and have not reoccurred multiple times using the same breakpoint regions; a pattern observed in other species[35,36]. However, it should be noted that for the inversion on Chr6, the distal breakpoint slightly deviated from this common observation for one of the samples,

**Table 1 | Genome statistics for the CS10 and BS3 de novo genome assemblies in comparison with the current reference assembly**

| Samples | Genome size (Mb) | BUSCO[64] scores (%) | No. contigs | N50 (kb) | N's per 100 kb |
|---------|------------------|----------------------|-------------|----------|----------------|
| CS10 | 773.1 | 90.7 | 3997 | 452.8 | 0 |
|  | 756.6 | 90.3 | 3228 | 472.4 | 0 |
| BS3 | 779.2 | 92.4 | 3331 | 680.2 | 0 |
|  | 758.5 | 92.1 | 2725 | 661.8 | 0 |
| Reference | 725.7 | 87.4 | 26 | 30,022.5 | 110.8 |
|  | 786.3 | 93.3 | 1697 | 29,845.7 | 118.4 |

Hap1 and hap2 are the two haplotype assemblies for the PacBio samples and their statistics are shown on the top and bottom row for each sample, respectively. In the case of the reference, the top row indicates assembly with scaffolded chromosomes, and the bottom row represents assembly with unplaced scaffolds in addition to the scaffolded chromosomes.

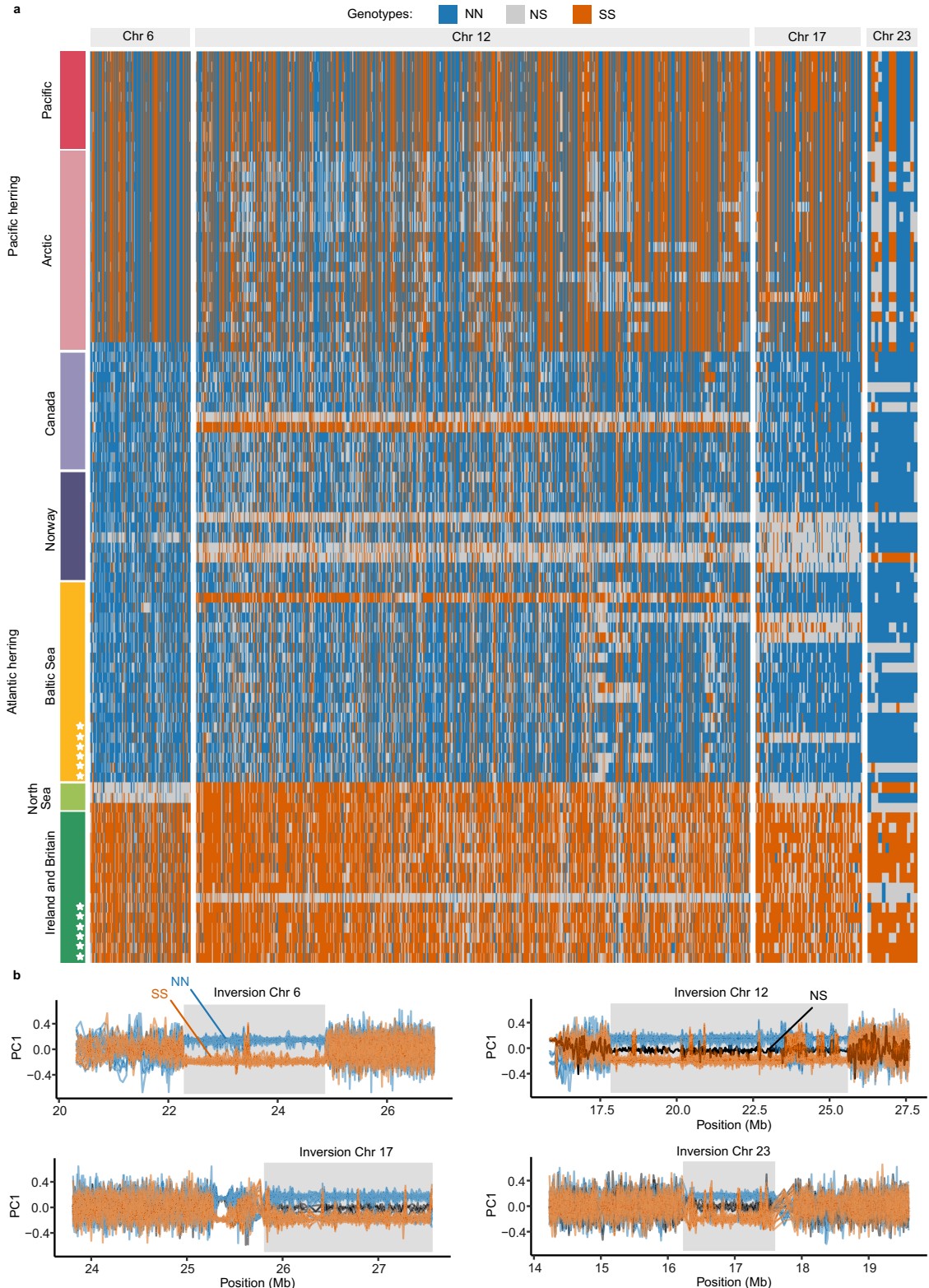

**Fig. 1 | Individual genotypes at the four inversions. a** Heatmap of genotypes of 91 individuals at highly differentiated SNPs between northern and southern populations of Atlantic herring[31] that overlap with the inversions on chromosomes 6, 12, 17, and 23. Each row represents the genotypes of one individual, and each column represents a SNP position. Colored bars represent Atlantic herring or Pacific herring populations. Stars represent individuals from the PacBio sample set. The "Arctic" block includes individuals from the White Sea, Pechora Sea and Balsfjord, the latter a fjord in the North-Atlantic Ocean harboring a Pacific-Atlantic herring hybrid

population[33]. Genotypes are color-coded depending on their homozygosity or heterozygosity for *N* and *S* alleles. Genotypes are polarized assuming that the *S* allele is in high frequency (>0.5) in Ireland and Britain. **b** Sliding window PCA analysis across inversion regions (sliding windows of 200 SNPs). Each line represents one of 35 individuals from the Baltic and Celtic Sea (dark green and yellow individuals from panel **a**). Individuals are color-coded according to their genotype at the inversion: blue if homozygous for the *N* allele, orange if homozygous for the *S* allele and black if heterozygous.

being 500 kb further along the chromosome (Supplementary Fig. 2, Table 2). This could be due to either a different distal breakpoint or a secondary inversion, but more samples are needed to confirm these possibilities. We re-evaluated the breakpoints obtained in our previous study[31] for Chr6 and Chr17 inversions and found 1–3 kb shifts at two positions, mainly because some gaps in the reference assembly are closed in the new PacBio assemblies (Table 2). Figure 2 reports the consensus breakpoint coordinates as those that occurred most frequently in the examined samples, which were Chr6:22,282,765-24,868,682, Chr12:17,826,318-25,603,093, Chr17:25,802,209-27,568,510, Chr23:16,225,343-17,604,279. None of these breakpoints disrupted any coding sequence (Fig. 2; the list of genes around the breakpoints is provided in Supplementary Data 1). The inversions were further confirmed by the alignments of $N$ and $S$ allele scaffolds constructed using PacBio contigs and optical mapping data (Supplementary Fig. 3). The breakpoint coordinates on $N$ and $S$ allele scaffolds were determined by noting the coordinates where the scaffolds change orientation in the sequence alignment dot plot (Supplementary Table 3).

## Structural variations at the breakpoint regions

Leveraging the long PacBio contigs spanning the inversion breakpoints and the optical genome mapping data, we studied structural variants (SVs) and repeats surrounding the inversion breakpoints in each haplotype, which could have played a role in the formation of the inversions. The sequence alignments of $N$ and $S$ alleles near the breakpoints indicated that the breakpoints for three of the four inversions were flanked by inverted duplications ranging from 8 to 60 kb in size and contained one or no gene (Figs. 2 and 3; Supplementary Data 1); the breakpoint structure for the inversion on Chr23 is difficult to interpret. Further examination of these inverted duplications (Supplementary Table 4) revealed sequence identities between the proximal and distal copies in the range 85–99%, and only core regions (ranging from 7 to 21 kb in length) share high sequence identity (ranging from 95 to 99%). However, such highly identical sequences were not found on the Chr17 $S$ allele (Supplementary Table 4). Comparison of inverted duplications among different inversions revealed no sequence similarity. In addition, other types of SVs, like indels, palindromes and duplications, were also enriched near the breakpoints (Supplementary Table 5).

We also studied SVs near and inside inversion haplotypes using genome graphs (Supplementary Fig. 5), which corroborated the complexity of breakpoint regions already apparent in the dot plot analysis (Fig. 3). For instance, distal breakpoints of all inversions were divergent among individuals, revealing the existence of non-shared structural variants (Supplementary Figs. 4, 5; Supplementary Table 5). In particular, the Chr17 and Chr23 breakpoints were the most complex. The distal breakpoint of Chr17 coincided with a telomeric sequence that varies in length (0–300 kb) outside the breakpoint and that is misaligned in the genome graph (Supplementary Fig. 5). The Chr23 inversion breakpoints were the most divergent across individuals, revealing the existence of a long breakpoint region with many structural variants. This complexity suggests that, after the formation of an inversion, there could be an accumulation of structural variants around the breakpoints, in this case, not associated with any particular inversion allele and that may be evolving neutrally. The genome graphs (Supplementary Fig. 5) also revealed the existence of structural variants inside inversions, in particular for Chr12, 17, and 23, while Chr6 alignments revealed higher similarity among haplotypes.

## Identification of ancestral haplotypes using European sprat as an outgroup

Previous reports have shown that estimates of nucleotide diversity ($\pi$) for the $N$ and $S$ haplotypes are similar for all four inversions[31] and, thus,

cannot be used to determine the ancestral vs. derived state of the haplotypes. To overcome this obstacle, we here use the recently released high-quality reference genome of European sprat (*Sprattus sprattus*) *Darwin Tree of Life*. https://portal.darwintreeoflife.org/data/root/details/Sprattus sprattus. as an appropriate outgroup species[37,38], and by such determine which of the inversion haplotypes represent the ancestral vs. derived state. We find a high degree of conserved synteny between Atlantic herring and European sprat, but the chromosome number differs, 26 vs. 20, respectively (Supplementary Fig. 6a), and there are many interchromosomal rearrangements between the two species. However, long stretches of chromosomes, including those containing inversions in herring, align with sequence identities between 0.25 and 0.75 (Supplementary Fig. 6b, c). We investigated the inversion regions in detail using dot plots comparing the sprat sequence with the $N$ and $S$ alleles. The linear orientation of the alignment of sprat sequence to the $N$ and $S$ alleles suggests that $S$ is the ancestral haplotype for the inversions on Chr6 and 12; while $N$ is the ancestral haplotype for the inversions on Chr17 and 23 (Fig. 4). However, results for Chr23 should be treated with caution as the alignment was fragmented.

## Timing of the origin of inversion haplotypes

We used short-read sequence data from Atlantic and Pacific herring individuals mapped to the Atlantic herring reference genome to study the origin and subsequent evolution of the inversions. We first studied the genome-wide evolutionary history of the two *Clupea* sister species, Atlantic and Pacific herring, by either (1) using a concatenated alignment of 15,471 genes (~114 Mb with no missing data) including the European sprat to generate a rooted tree, or (2) by using a longer ~346 Mb genome-wide alignment with no missing data containing only *Clupea* individuals, to more confidently infer intraspecific relationships. Atlantic and Pacific herring formed well-supported monophyletic sister clades (Fig. 5a, b; Supplementary Fig. 7). Maximum likelihood trees of all four inversion haplotypes using data from multiple herring populations revealed a similar evolutionary history as the genome-wide species tree (Fig. 5; Supplementary Fig. 8). In rooted and unrooted trees, we found a split of all Atlantic herring individuals from the Pacific herring, followed by a split between reciprocal homozygotes of each inversion allele, with heterozygotes placed between these two clusters (Fig. 5c–f; Supplementary Fig. 8), suggesting that the inversions originated after the split between Atlantic and Pacific herring. The $S$ cluster is constituted by all individuals originating from Britain and Ireland and part of the North Sea individuals in all four trees, whereas the $N$ cluster is mostly constituted by Baltic Sea, Norwegian and Canadian herring. The coincidence of the phylogenetic relationship between $N$ and $S$ alleles and the geographic distribution of the individuals is in line with previous results that suggest that the $S$ alleles at each inversion tend to occur at high frequency in warmer waters, particularly around Britain and Ireland, whereas $N$ alleles occur at high frequency in colder waters in the north[31,39].

We used net nucleotide diversity ($d_a$) between the $N$ and $S$ homozygotes to estimate divergence among inversion haplotypes, relative to the divergence of Atlantic and Pacific herring (Fig. 5). Divergence times ranged from 1.51 million years (MY) (Chr 6) to 2.20 MY (Chr 17), which are more recent divergences than the one estimated for Atlantic and Pacific herring in our dataset (3.30 MY). Given that $d_{xy}$ values between $N$ and $S$ alleles were lower but close to $d_{xy}$ between Atlantic and Pacific herring across inversion regions (Supplementary Fig. 9a), and given the possibility of recombination among inversion haplotypes (Fig. 1; Supplementary Fig. 5 and results below)[29], it is possible that our estimated divergence times are underestimations, suggesting that the inversions are old polymorphisms that could have originated shortly after the split between the two *Clupea* species.

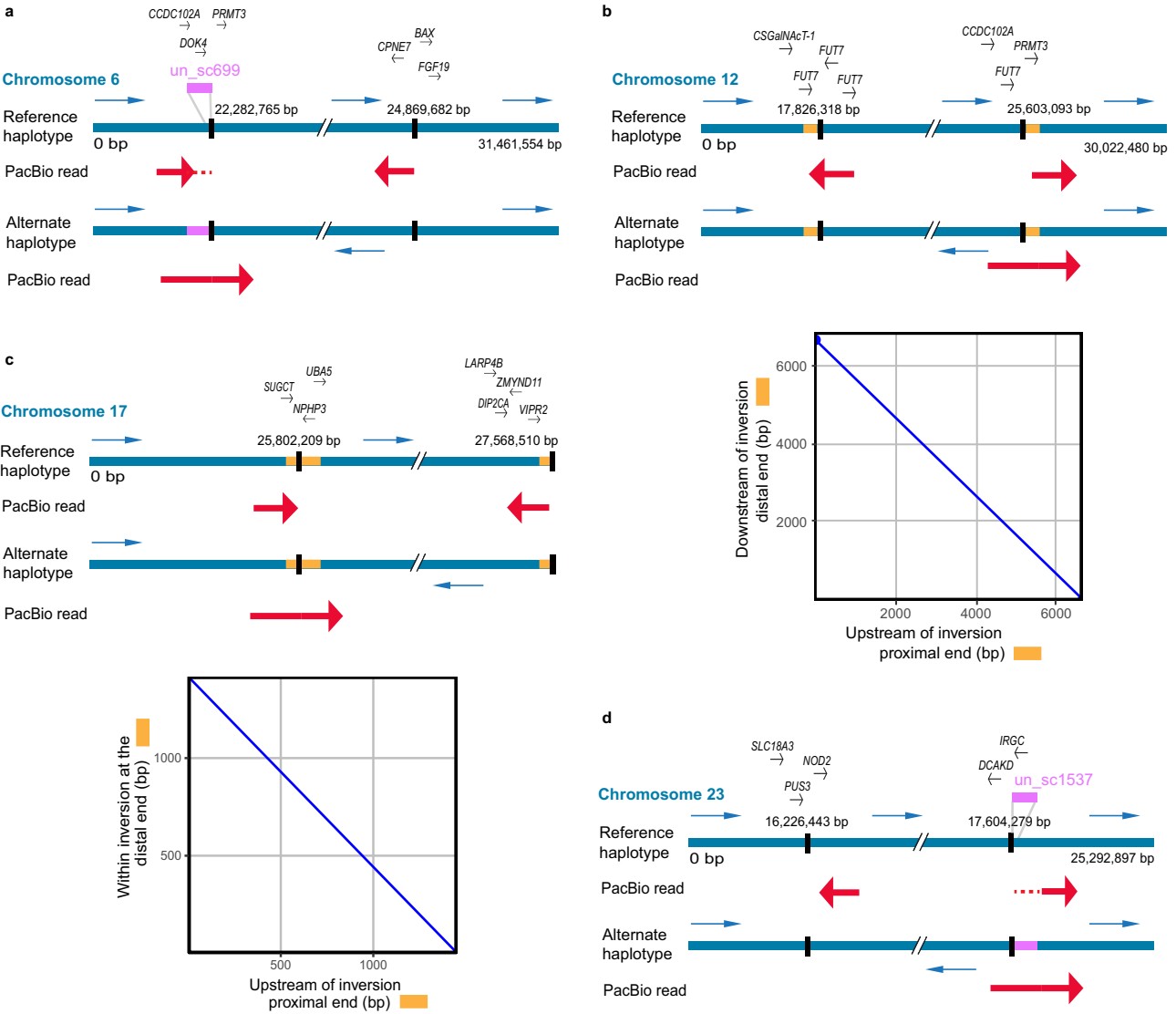

**Fig. 2 | Alignment of single PacBio reads to the inversion breakpoints on chromosomes 6, 12, 17, and 23.** The reference haplotype represents the sequence from the reference assembly, while the assembly of the alternate haplotype is shown below. Blue arrows represent the orientation of sequences in the two haplotypes. The breakpoints are indicated as vertical black lines. Alignments of single PacBio HiFi reads to the reference and alternate haplotypes are indicated with thick red arrows. Genes near the breakpoints are indicated above the reference sequence and described in Supplementary Data 1. **a** Chromosome 6 (modified from ref. 31). A part of the HiFi read maps to unplaced_scaffold (un_sc) 699 (indicated by red dotted lines) as the reference assembly is not scaffolded at this region. **b** Chromosome 12. **c** Chromosome 17 (modified from ref. 31). **d** Chromosome 23. A part of the HiFi read maps to un_sc 1537 (indicated by red dotted lines) as the reference assembly is not scaffolded at this region. The dot plots below the read alignments for chromosomes 12 and 17 compare inverted duplications flanking the breakpoints, which are also indicated as yellow boxes on the inversion haplotypes. The length of the inverted duplications on chromosomes 12 and 17 are 8 kb and 3 kb, respectively.

## The evolutionary history of inversion haplotypes

We explored the evolutionary history of the four inversions using homozygous individuals from the Baltic and Celtic Sea (total $n$ = 35; Fig. 1) using short-read data mapped to the Atlantic herring reference genome. We calculated sequence differentiation ($F_{ST}$), nucleotide diversity ($\pi$), and linkage disequilibrium (LD) measured as $R^2$ across the inversions and their flanking region (Fig. 6, Supplementary Fig. 10). The inversion regions showed strong differentiation between $N$ and $S$ homozygotes (high $F_{ST}$) which is in sharp contrast with the flanking regions (low $F_{ST}$). An exception to this is a region proximal to the Chr17 inversion breakpoint. A careful inspection of our PacBio data showed that this is not part of the inversion and must be a sequence polymorphism in very strong LD with the inversion polymorphism (Supplementary Fig. 11). The LD across all four inversions was strong, particularly for Chr6 and Chr12 inversions. Nucleotide diversity for all

four inversions showed significant differences between haplotypes and genome-wide averages in certain cases, but $\pi$ values of all inversions are within the genome-wide distribution of $\pi$ (Fig. 6) The nucleotide diversity of inversion alleles representing the derived state is not lower than for those representing the ancestral state (Fig. 6), as only the Chr 12 inversion showed a significantly reduced diversity in the derived haplotype ($P$ < 0.001 for Chr12) as expected, while Chr17 and Chr23 inversions showed higher diversity in the derived haplotype ($P$ ≪ 0.001 for Chr17 and $P$ < 0.01 for Chr23) (Fig. 6). These results are consistent with the old age of the inversion polymorphisms (Fig. 5) exceeding the coalescence time for neutral alleles in Atlantic herring.

Suppression of recombination between inversion haplotypes is expected to result in the accumulation of deleterious mutations and transposable elements (TEs) due to impaired purifying selection, as the inversion haplotypes have a reduced $Ne$ compared with the rest of the

**Table 2 | Inversion breakpoint coordinates on the reference assembly for chromosomes 6, 12, 17, and 23**

| Inversion | Samples | Proximal breakpoint | Distal breakpoint | Shift from the consensus breakpoint in bp (proximal/distal) |
|---|---|---|---|---|
| Chromosome 6[a,b] | CS2 | 22,282,765 | 25,427,801 | -/559,219 |
| | CS4 | 22,282,765 | 24,869,682 | -/1100 |
| | CS7 | 22,282,765 | 24,869,682 | -/1100 |
| | CS10 | 22,282,765 | 24,869,682 | -/1100 |
| Chromosome 12 | BS1 | 17,826,318 | 25,603,093 | -/- |
| | BS2 | 17,826,318 | 25,603,093 | -/- |
| | BS3 | 17,826,318 | 25,603,093 | -/- |
| | BS4 | 17,826,318 | 25,603,093 | -/- |
| | BS5 | 17,826,318 | 25,603,093 | -/- |
| | BS6 | 17,826,318 | 25,603,093 | -/- |
| Chromosome 17[a] | CS2 | 25,802,209 | 27,568,510 | -/- |
| | CS4 | 25,802,209 | 27,568,510 | -/- |
| | CS5 | 25,802,212 | 27,568,510 | 3/- |
| | CS7 | 25,802,212 | 27,568,510 | 3/- |
| | CS8 | 25,802,209 | 27,568,510 | -/- |
| | CS10 | 25,802,209 | 27,568,510 | -/- |
| Chromosome 23 | BS1 | 16,225,343 | 17,604,279 | -/- |
| | BS2 | 16,216,922 | 17,604,291 | 8421/12 |
| | BS3 | 16,225,343 | 17,604,291 | -/12 |
| | BS4 | 16,225,343 | 17,604,279 | -/- |
| | BS5 | 16,225,343 | 17,604,277 | -/2 |
| | BS6 | 16,226,443 | 17,603,173 | 1100/1106 |

The last column presents shift from the consensus breakpoints, which are Chr6:22,282,765-24,868,582, Chr12:17,826,318-25,603,093, Chr17:25,802,2019-27,568,510, Chr23:16,225,343-17,604,279.

[a]Chromosome 6 and 17 breakpoints show 1–3 kb shifts from the previously reported breakpoints[31].

[b]The samples with ID CS5 and CS8 had no aligned reads spanning the breakpoints of chromosome 6 inversion because the reference sequence at the breakpoint had gaps, SVs, and mis-assembled sequence.

genome[22]. To test this, we compared the number of non-synonymous substitutions per non-synonymous site ($d_N$) to the number of synonymous substitutions per synonymous site ($d_S$), or $d_N/d_S$ and site frequency spectrum of non-synonymous and synonymous mutations for genes within the inversions to the genome average, using the European sprat as an outgroup species. We found no significant difference in $d_N/d_S$ for any inversion allele and the genome-wide distribution ($P > 0.05$, two-sided $t$-test, Fig. 7a). Further, the site frequency spectrum of $N$ and $S$ homozygotes were similar to each other (Fig. 7b) and to that of the genome-wide estimate (Supplementary Fig. 12), where polymorphic synonymous positions are always the most abundant class, suggesting that low-frequency non-synonymous mutations are being effectively purged from inversion haplotypes. Further, we compared the $d_N/d_S$ ratio for the $N$ and $S$ alleles at each locus in an attempt to find genes that may show accelerated protein evolution as part of the evolution of these adaptive haplotypes. However, the ratios were remarkably similar in pairwise comparisons, with only a few genes showing a minor difference in $d_N/d_S$ (Supplementary Fig. 13).

Finally, we compared TE abundance between $N$ and $S$ haplotypes, as a proxy of mutational load, which revealed non-significant difference between haplotypes and a lower TE content in Chr6, Chr12, and Chr17 inversions compared to the rest of the genome (Fig. 7c). Taken together, the data on $d_N/d_S$ and on TE content did not indicate increased genetic load for alleles at any of the four inversion polymorphisms.

## Evidence of allelic exchange between inversion haplotypes

To visualize genetic exchange between inversion haplotypes (gene flux), we constructed a deltaAlleleFrequency' (dAF') metric that measures the degree of allele sharing between haplotypes (see "Methods"). dAF' = 1.0 means that there is a maximum dAF given the frequencies of sequence variants among haplotypes, while dAF' = 0 means that sequence variants have the same frequencies among the two haplotype groups. All sequence variants within an inversion will show dAF' = 1 if there has been no gene flux and the same mutation has not occurred on both haplotypes. This analysis documents extensive allele sharing at all four loci, and in particular for Chr12 and 17 (Fig. 8), because if there had been no gene flux, all SNPs within the inversion would have dAF' ~ 1 (colored blue in Fig. 8). The result is consistent with our previous analysis of allele sharing for the Chr12 inversion[40]. The region between Chr12: 23.0–23.5 Mb, with a particularly high incidence of sequence variants with low dAF' values correspond to an interval where we have noted evidence for genetic recombination between the $N$ and $S$ alleles[41], where we see a drop in $F_{ST}$ between haplotypes (Fig. 6), an excess of heterozygous genotypes in Baltic Sea individuals (Fig. 1a), and reversal of PCA loadings in $SS$ and $NN$ individuals (Fig. 1b). This analysis also confirms the extreme sequence divergence between $N$ and $S$ homozygotes in a flanking region outside the inversion for the Chr17 inversion. Extremely differentiated SNPs (dAF > 0.95) were not enriched for non-synonymous mutations (Supplementary Table 6, a list of genes with non-synonymous mutations is in Supplementary Data 2).

## Discussion

In this study, we leverage the power of long-read sequencing to confirm the presence of four large inversions in Atlantic herring that all show strong differentiation between populations from the northern versus southern part of the species distribution[31]. Our detailed analysis of the breakpoint regions indicated that ectopic recombination between inverted duplicates flanking the inversions might have been essential for their formation. By comparing our assemblies to long-read data for an outgroup species, European sprat, we determine that the $S$ arrangement is ancestral at Chr6 and Chr12, whereas $N$ is ancestral at Chr17 and Chr23. This differs from our previous prediction[31] as regards Chr17, which was only based on data from the reference assembly, which contain truncated sequence at the distal breakpoint, an issue which has now been resolved with the new PacBio data from 12 individuals. We also used extensive re-sequencing data from the Atlantic herring and its sister species, the Pacific herring, to study the evolutionary history of the inversions. Our phylogenetic analysis shows that the four inversion polymorphisms have been maintained for more than a million years but they most likely all occurred subsequent to the split between Atlantic and Pacific herring. Our population genetic analysis revealed no indication of accumulation of mutation load, despite high differentiation and strong suppressed recombination in the region. The signatures of genetic exchange between inversion haplotypes and their high frequency in different populations suggest that these inversions have been maintained for a long evolutionary period by divergent selection related to ecological adaptation.

### Inverted duplications present near inversion breakpoints

Here we characterized the breakpoint regions in detail, revealing the role of structural variation in the origin of the inversions and their subsequent evolution. We found similar chromosomal breakpoints in all haplotypes (Table 2), suggesting that each of the four inversions has a single origin. Phylogenetic trees of the inversion haplotypes also support this result, since individuals cluster by their genotype at the inversion ($NN$ or $SS$), rather than geographic location (Fig. 5). The inversion breakpoints were surrounded by multiple SVs (Fig. 3), of which inverted duplications would potentially be playing an important

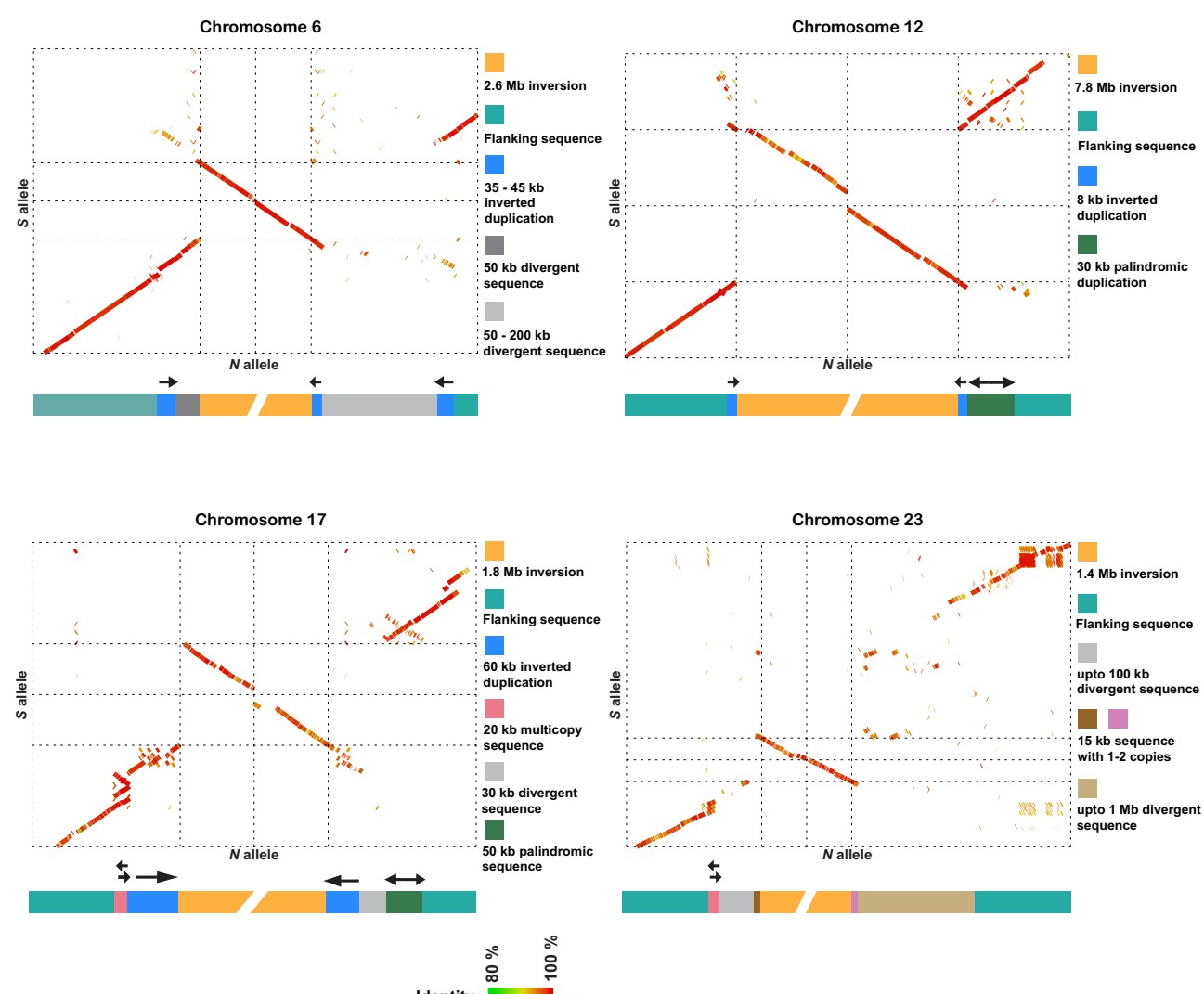

**Fig. 3 | Sequence alignments and models at the breakpoint regions for the *N* and *S* inversion allele scaffolds.** The green to red color gradient in dot plots represents percent sequence identity from 80 to 100%. The models representing SVs surrounding the inversions are below the dot plots. Chr6: 400 kb sequence includes 300 kb outside inversion and 100 kb inside inversion for both proximal and distal regions. Chr12: 200 kb sequence includes 100 kb outside inversion and 100 kb inside inversion for both proximal and distal regions. Chr17: 400 kb sequence includes 300 kb outside inversion and 100 kb inside inversion for both proximal and distal regions. Chr23: 400 kb proximal. 600 kb distal sequence includes 500 kb outside inversion and 100 kb inside inversion. The plots are based on the CS10_hap1 and BS3_hap2 genome assemblies.

role in the formation of inversions by nonallelic homologous recombination (NAHR) (also referred to as ectopic recombination). This mechanism of inversion formation is commonly accepted[16] and reported in a few species of *Drosophila* and eutherian mammals[25,40,42–44], but the homologous sequences at base pair resolution leading to NAHR are not studied due to the complexity of the region. We analyzed these sequences using *N* and *S* inversion scaffolds constructed using PacBio contigs and optical genome mapping and found 8–60 kb block of inverted duplicates with 85–99% sequence identity (Fig. 3), but on close inspection, only 7 to 21 kb sequence of these duplication pairs shared high sequence identity (95–99%). The absence of such high identity region in Chr17 *S* allele could be due to non-integrity of the genome assembly at this complex region (Supplementary Table 4). From multiple studies in humans, it is known that such genomic architecture with low-copy repeats ranging from 10 to 400 kb with ≥ 95% sequence identity constitute recombination hotspots causing chromosomal rearrangements mediated by different mechanisms including NAHR[16,45,46]. Our findings suggest that it is likely that the herring inversions were caused by ectopic recombination

between inverted duplications, similar to the complex genomic rearrangements responsible for genomic disorders in humans[16,45,46].

As we find no inverted duplications for the inversion on Chr23, it is possible that it has been formed by the alternative mechanism of double-strand staggered breaks. This mechanism is argued to be the most common mechanism in invertebrate species, where a single-stranded break is repaired by Nonhomologous DNA End Joining (NHEJ) and may result in inversion accompanied by duplication[14,15]. Nevertheless, NAHR seems to be the most common mechanism in vertebrates[25,47], and indeed our data supports a prevalent role of NAHR in the formation of at least three of the four inversions in Atlantic herring. The presence of flanking inverted duplication sequences increases the probability of recurrent inversions, an event termed as "inversion toggling", by breakpoint reusage[48]. All the breakpoints were surrounded by repeated, palindromic, and divergent sequences (Fig. 3; Supplementary Fig. 5), which could have been formed by a gene conversion process using the inverted duplicates flanking the inversion breakpoints. Such a process can further facilitate the formation of insertions and deletions, during which, double-stranded breaks, strand

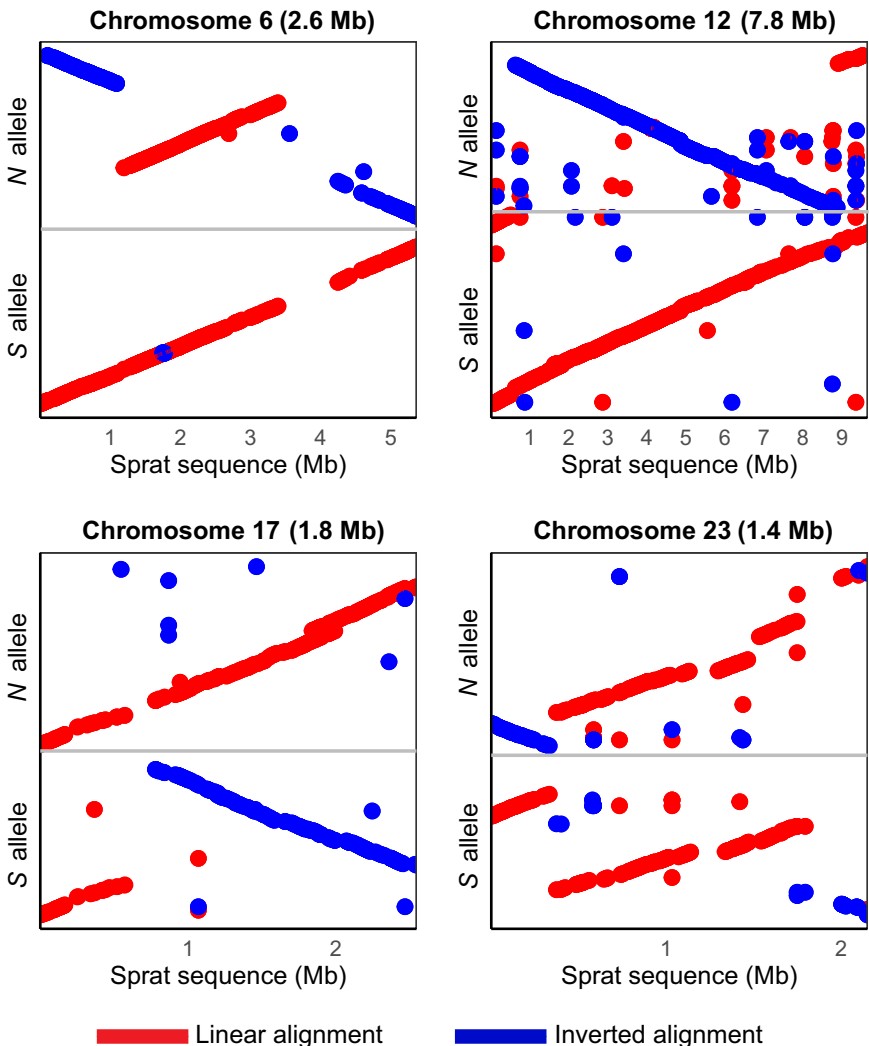

**Fig. 4 | Sequence alignment of Atlantic herring inversion alleles and contigs from the European sprat assembly spanning the inversion breakpoints for all four inversions.** Forward and reverse orientation of the alignments are represented by red and blue dots, respectively.

extension, and rejoining create even more duplicated sequences[40]. Notably, most of these SVs outside of inversions occur in non-genic regions. Presence of such divergent sequences around breakpoint might be responsible for restricting the gene flow at the breakpoints and thus maintaining the diversity among haplotypes, as peaks of divergence are common at the breakpoints of old inversions[12,13]. SVs inside the inversions showed a strong correlation with the inversion haplotype, suggesting that inversion haplotypes are evolving under strong selection.

Overall, the individual PacBio genome assemblies were crucial to understand the nature of inverted duplications and other SVs, and it was not sufficient to just align the PacBio reads to the reference genome assembly. In the case of Chr17, the read alignment (Fig. 2) gave an incomplete view of the positioning of the inverted duplications as the sequence past the distal end of the inversion is missing in the reference assembly (PacBio assemblies contain sequence ranging from 0 to 300 kb in size past the distal point, Supplementary Figs. 4, 5).

### The evolutionary history of the inversions is marked by events of gene flux

Our phylogenomic analysis revealed that the four Atlantic herring inversions originated after the split from its sister species, the Pacific herring, between ~1.5 and ~2.2 MYA (Fig. 5), or 2.5 and $3.7 \times 10^5$ generations ago, considering a generation time of six years for Atlantic

herring[49]. Given that ancestral $N_e$ for Atlantic herring has been estimated at $4 \times 10^5$ (ref. 32), inversions are of similar age to the coalescent time for neutral alleles (age ~ $N_e$), which should be enough time for exchange of variants between inversions by recombination (gene flux). In fact, our population genetics and dAF' analyses document gene flux in all four inversions and recombination through double crossover in the Chr6 (at ~23.5 Mb) and Chr12 (at 23.25–24.0 Mb) inversions (Figs. 1, 6, 8, Supplementary Fig. 5). This is also in line with previous evidence for gene flux in the Chr12 inversion[41]. Due to gene flux, it is possible that our age estimates are underestimated.

It is expected that as inversions reach mutation-drift-flux equilibrium, gene flux erodes the divergence between haplotypes for neutral polymorphisms at the center but not at the breakpoints, resulting in a U-shaped pattern for divergence and differentiation, as reported in some *Drosophila* species[12,13]. Strong signals of such erosion were not observed in our data, since all four inversions showed high $F_{ST}$, $d_{xy}$, and strong LD across the inversion (Fig. 6), a pattern consistent with a situation where polymorphisms under selection are distributed across the inversion as previously reported in *Drosophila*[12,50,51]. A similar pattern was also reported for old inversions present in Atlantic cod[26]. However, $F_{ST}$ and LD were relatively weaker for Chr17 and 23 (Fig. 6), along with a slight reduction of $d_{xy}$ at the center compared to the breakpoints (Supplementary Fig. 9), thus weakly supporting the expectations for neutral polymorphisms dominating at the center of

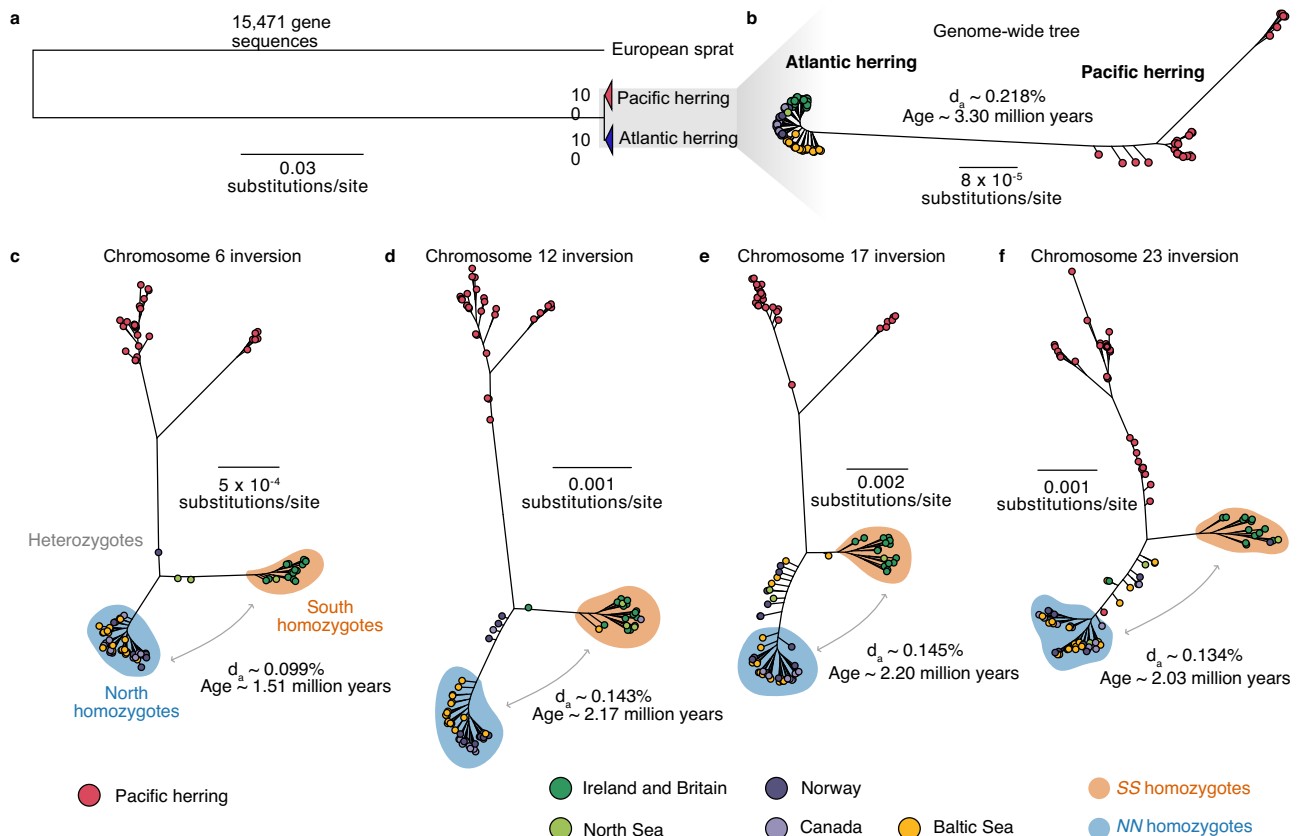

**Fig. 5 | The evolutionary history of chromosomal rearrangements in Atlantic herring. a** Maximum likelihood tree of 15,471 concatenated gene sequences from 30 Pacific and 61 Atlantic herring, using the European sprat as an outgroup; **b** Maximum likelihood tree of 345,966,161 bp concatenated genome-wide positions with no missing data using the same 91 individuals; **c**–**f** Maximum likelihood trees of the inversion regions (alignments allowing 50% missing data). Estimated net nucleotide divergence ($d_a$) and divergence times in million years between Atlantic and Pacific herring (**b**) and between $N$ and $S$ homozygotes (**c**–**f**) are indicated. In trees (**b**–**f**), Atlantic herring individuals are color-coded depending on the population of origin.

these inversions. The Chr6 inversion also showed high divergence at breakpoints, but the pattern continued after the distal breakpoint, which can be attributed to its presence in a high diversity region of the genome (Supplementary Fig. 9). Together, the patterns of differentiation and linkage disequilibrium among inversion haplotypes and allele sharing among Atlantic herring inversions show that gene flux contributes to the evolution of the four inversions, but it is not strong enough to completely homogenize differentiation between chromosomal arrangements given their evolutionary age, and it is most likely counteracted by divergent selection for sequence polymorphisms contributing to ecological adaptation, i.e., natural selection is removing transferred gene variants that are maladaptive on the recipient inversion haplotype.

The nucleotide diversity shows variable patterns between derived and ancestral haplotypes across the four inversions (Fig. 6), where Chr6 and Chr12 inversions have lower diversity in the derived haplotype, as expected, while Chr17 and Chr23 have lower diversity in the ancestral haplotype. Higher diversity in the derived haplotype deviates from the expectations that the formation of an inversion leads to a strong loss of diversity in the derived haplotype when compared to the ancestral one[10,12,13]. However, such observation is not uncommon in natural systems[26] and could be explained by the recovery of nucleotide diversity by the derived haplotype after the initial bottleneck when this haplotype is maintained at high frequency and in natural populations with large $N_e$, such as Atlantic herring populations. Furthermore, gene flux between inversions could contribute to increase of nucleotide diversity of inverted haplotypes over time[13].

## Atlantic herring inversions have evolved due to divergent selection and show no significant mutational load

The four inversions in Atlantic herring studied here show highly significant genetic differentiation among subpopulations of Atlantic herring, implying a key role in local adaptation. The general pattern is that the allele named *Southern* dominates in the southern part of the species distribution while the *Northern* allele dominates in the north[31]. We calculated the population migration rate ($Nm$) according to Slatkin[52] among populations of Atlantic and Baltic herring and noted that $Nm$ is much higher than 1.0 and thus sufficiently strong to homogenize divergence among populations (Supplementary Fig. 14). The level of differentiation between inversion haplotypes is far above genome-wide average $F_{ST}$ (Fig. 6; Supplementary Fig. 14) supporting our interpretation that inversion haplotypes are maintained by divergent selection. This is consistent with our previous analysis demonstrating that the distribution of $F_{ST}$ among herring populations deviates significantly from the one expected for neutral polymorphisms under a genetic drift model[53]. Overdominance can be an important mechanism for the maintenance of inversion polymorphisms[54]. However, we find no indication that overdominance is important for the herring inversions because at all four loci both haplotypes reach fixation or close to fixation in some populations. In populations where both haplotypes segregate, we find no significant deviation from Hardy-Weinberg equilibrium (Supplementary Table 7). It is possible that the inversions per se initially provided a phenotypic effect contributing to adaptation as suggested to be a mechanism for the establishment of inversion polymorphisms[54]. Our data imply that, if these inversions have a direct functional impact, it must be through

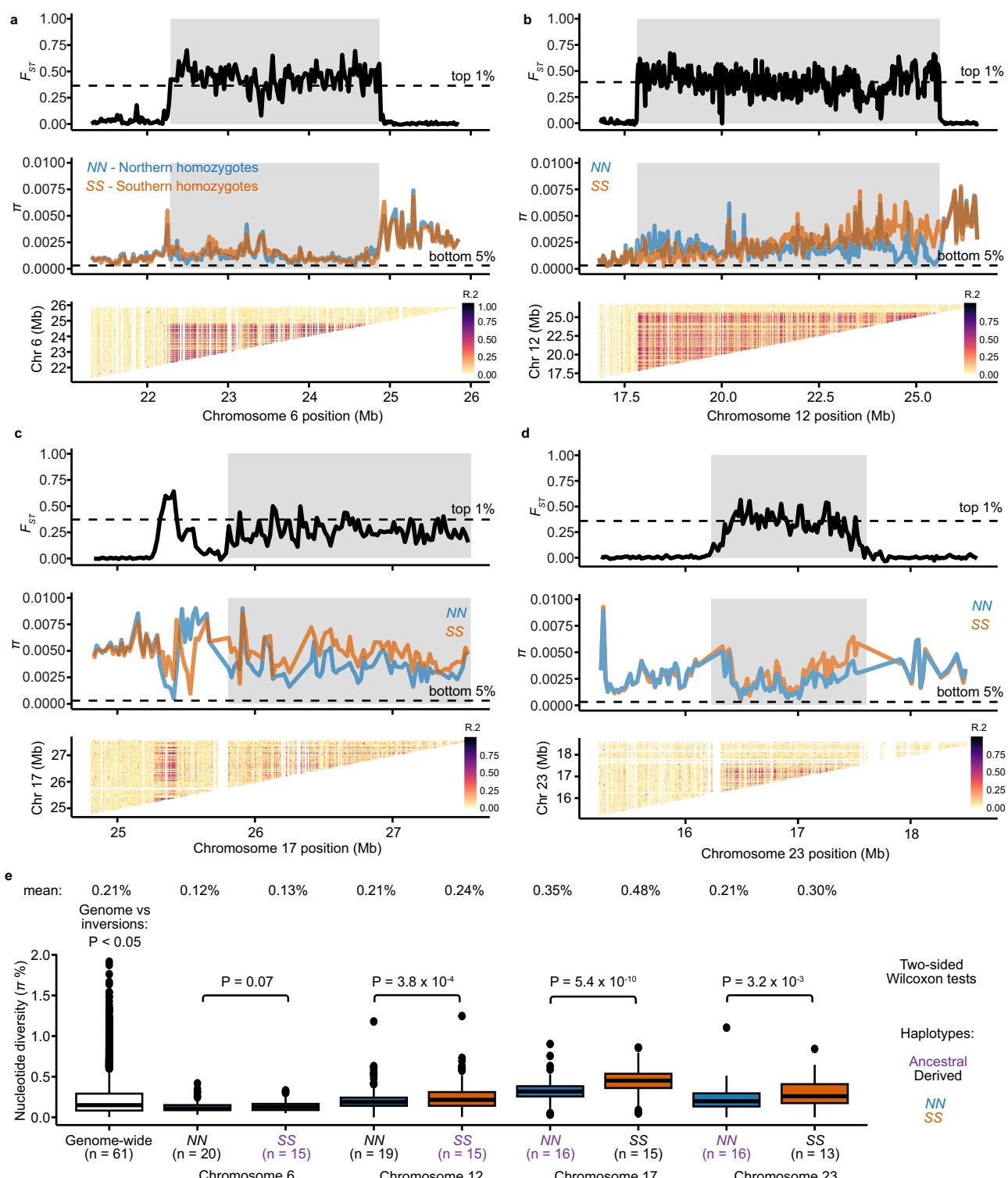

**Fig. 6 | Differentiation, diversity, and recombination patterns in inversion regions in Atlantic herring on Chr6, Chr12, Chr17, and Chr23. a** Chr6, **b** Chr12, **c** Chr17, and **d** Chr23. Distribution of $F_{ST}$ and nucleotide diversity ($\pi$) for $N$ and $S$ homozygotes are displayed in sliding windows of 20 kb across inversion regions (shaded gray boxes). Linkage disequilibrium is represented as $R^2$ among genotypes in the inversion region. We thinned SNPs positions to perform this calculation (see "Methods"). **e** Comparison of the distribution of nucleotide diversity ($\pi$) calculated in sliding windows of 20 kb for each inversion haplotype and genome-wide. Sample size is indicated below each boxplot. Nucleotide diversity for each group is displayed above the boxplots, and ancestral alleles are highlighted in purple. Boxplots present the median, 25th and 75th percentiles of the distribution. The results are based on short-read data mapped to the Atlantic herring reference genome.

effects on gene regulation, as no coding sequences have been disrupted at any of the breakpoints (Fig. 2, Supplementary Fig. 3, Supplementary Data 1). Alternatively, the inversion may have captured a combination of favorable alleles at two or more loci, acting like a

supergene[11]. After their origin more than a million years ago, the alternative inversion haplotypes most likely have accumulated additional mutations contributing to fitness, and the haplotypes have been maintained by divergent selection even in the presence of gene flow

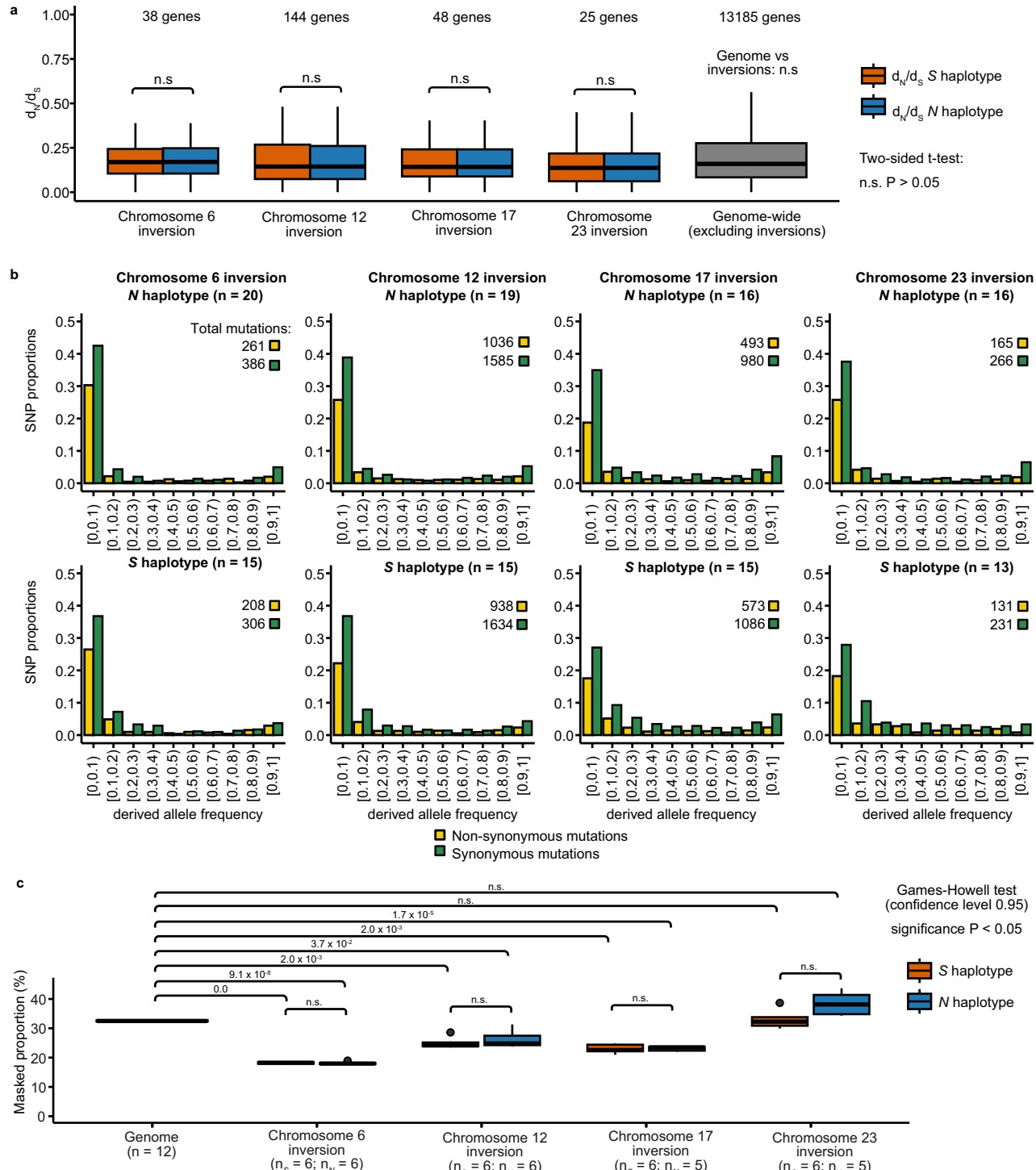

**Fig. 7 | Lack of mutational load in *N* and *S* inversion haplotypes. a** Distribution of $d_N/d_S$ ratios for *N* and *S* alleles of genes overlapping with the inversions on chromosomes 6, 12, 17, and 23 compared with the genome-wide $d_N/d_S$ distribution. A two-sided *t*-test revealed no significant differences in the distribution of $d_N/d_S$ values between *N* and *S* alleles, or between inversion haplotypes and the genome. **b** Site frequency spectra of derived non-synonymous and synonymous mutations for genes inside the inversions, for each inversion haplotype. The total number of mutations in each category is displayed above the graph. **c** Proportion of transposable elements (TEs) for the entire genome and for each inversion haplotype. A Games-Howell non-parametric two-tailed test was performed to test differences between the proportion of TEs between each allele and the genome (n.s. = non-significant). **a**, **c** Boxplots represent the median, 25th and 75th percentiles of distributions.

between populations and gene flux between haplotypes. Recently, divergent selection associated with local adaptation to contrasting environments has been similarly invoked to explain the maintenance of inversion polymorphisms across deer mice[25,55], redpolls[24] and Atlantic salmon[27].

It is generally assumed that inversion polymorphisms lead to the accumulation of genetic load due to suppression of recombination and the reduced $N_e$ for inversion haplotypes compared with other parts of the genome[11]. Genetic load associated with inversion polymorphisms has been well documented in, for instance, *Drosophila*[19],

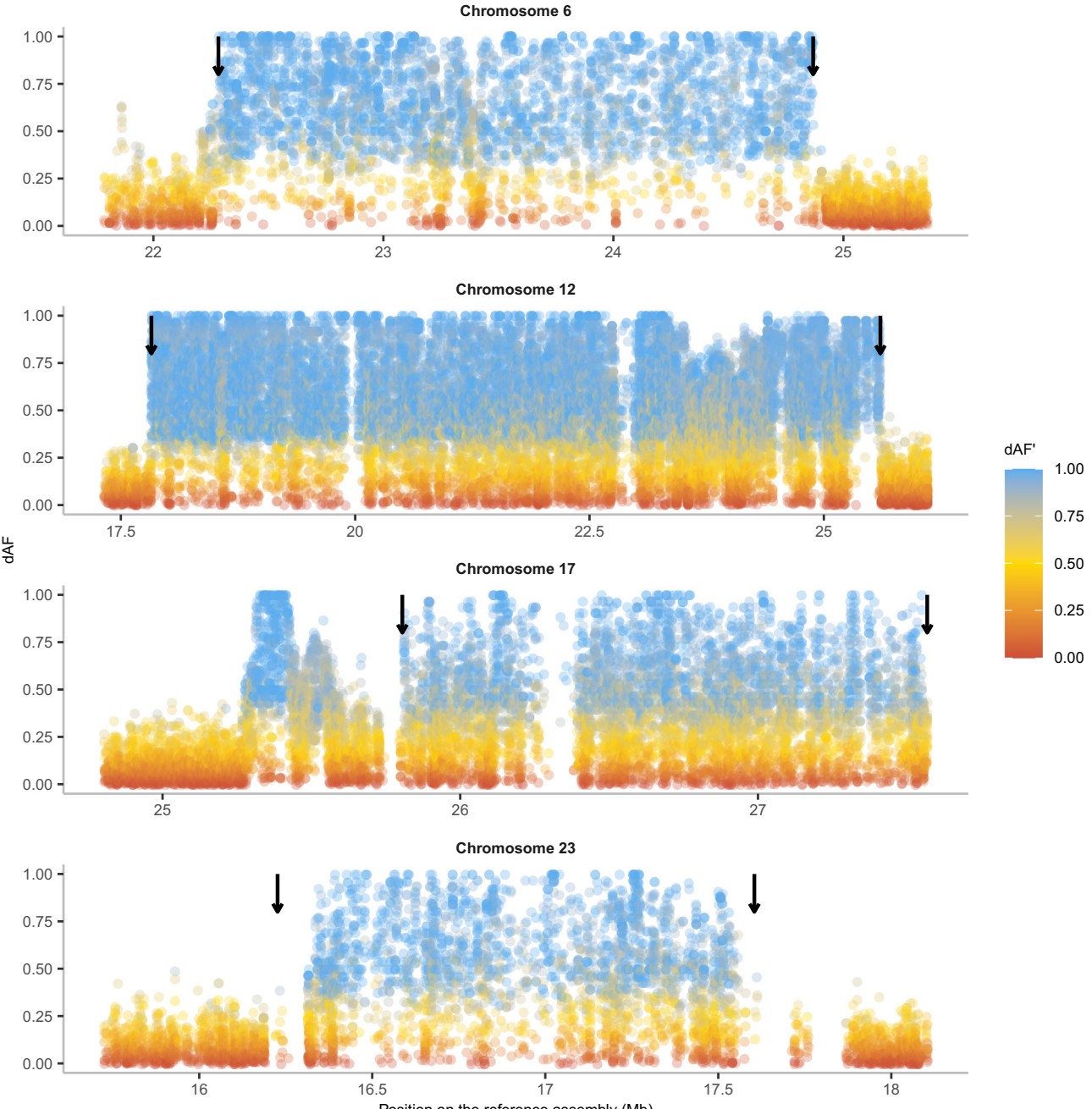

**Fig. 8 | dAF between *N* and *S* populations homozygotes for four inversions, colored by dAF'.** Arrows indicate the inversion breakpoints. The blank regions on chromosomes 12, 17, and 23 represent regions where SNPs are not called due to the complexity of the genomic regions with many repeats and indels. The results are based on short-read data mapped to the Atlantic herring reference genome.

seaweed flies[20,56], and *Heliconius* butterflies[7]. However, we find no evidence for genetic load associated with the four inversions (Fig. 7). All *N* and *S* inversion haplotypes are non-lethal and found at high frequencies in different populations. As Atlantic herring is an extremely abundant species, census population size in the order of $10^{12}$ (ref. 50), we expect the existence of billions of homozygotes for each haplotype in which recombination occurs at a normal rate. Thus, the lack of genetic load is consistent with the presence of effective purifying selection at these loci, as we find no signature of suppressed recombination causing notable linkage disequilibrium within inversion classes, which could hamper effective purging of deleterious mutations (Supplementary Fig. 10). A similar lack of genetic load has previously been reported for other inversion polymorphisms associated

with local adaptation in Atlantic cod[26], deer mice[25] and sunflower[57]. The results suggest that accumulation of genetic load does not occur for supergenes that are fully viable in the homozygous state and when both homozygotes are common in at least some populations, as is the case for supergenes associated with local adaptation.

Our study sheds new light on the mechanisms that contribute to the origin and govern the evolutionary history of inversions in natural populations. Leveraging the power of long-read sequencing using multiple individuals, we deduced accurate inversion breakpoints of all four inversions and found that none of the breakpoints disrupt the coding sequence of any of the genes. We found that the majority of the inversion breakpoints are flanked by inverted duplications, possibly responsible for the origin of inversions by ectopic recombination

between these sequences. The resolution provided by our population-level long-read dataset also reveals that the inversion breakpoints are highly enriched for structural variants and multiple structural variants are also present within the inversions, making the inversion haplotypes highly polymorphic. Our phylogenetic and population level analyses also support that inversion polymorphisms can be maintained by divergent selection for alternatively adaptive haplotypes in the face of strong gene flow in a species with massive population sizes. We find no evidence for the accumulation of mutational load or that over-dominance is important for the maintenance of inversion poly-morphisms in Atlantic herring, suggesting that the high $N_e$ of $N$ and $S$ haplotypes combined with gene flux events should allow efficient purifying selection on both inversion alleles. Our work contributes to a better understanding of what evolutionary factors govern the main-tenance of inversion polymorphisms in natural populations, which is key to determine their role in adaptive evolution and speciation.

# Methods

## Long-read dataset and construction of PacBio genome assemblies

Atlantic and Baltic herring samples were collected from commercial fisheries practice. Testis samples from 12 Atlantic herring, six from the Celtic Sea (collected on November 11, 2019, at latitude N51°59′ and longitude W6°48′) and six from the Baltic Sea (collected on May 18, 2020 in Hästskär, at latitude N60°35′ and longitude E17°48′) were used, representing the populations with a high frequency of the Southern (*S*) and Northern (*N*) inversion alleles[31], respectively. Tissue was extracted on-site and immediately flash-frozen in liquid nitrogen. High molecular weight DNA was extracted using a Circulomics Nanobind Tissue Big DNA Kit (NB-900-701-001) and sized to 15–25 kb using Bioruptor (Diagenode, Denville, NJ, USA). Sequencing libraries were constructed according to the manufacturers' protocols, and each sample was sequenced on one PacBio Sequel II 8M SMRT Cell for 30 h in circular consensus sequencing mode to generate about 20 Gb of HiFi sequence data. Similar data from an outgroup species, the European sprat (*Sprattus sprattus*), was derived from an initiative to establish a reference genome for this species[58]. The quality of HiFi data for all samples was assessed using NanoPlot[59].

For the assembly construction, we tested two genome assem-blers, HiCanu (v2.0)[60] and hifiasm (v0.16.1-r375)[34], which are specifi-cally developed for building genome assemblies using PacBio HiFi data. Hifiasm separated the diploid genomes into primary (hap1) and secondary (hap2) haplotypes. To separate HiCanu diploid genomes, we used Purge_dups[61]. QUAST (v5.0.2)[62] was used to evaluate genome statistics of all assemblies. The presence of conserved orthologs was assessed by BUSCO (v5.beta) using the vertebrate database[63]. We noted that the secondary haplotype assemblies generated by HiCanu were more fragmented than its primary counterpart (Supplementary Table 2). Moreover, we observed that most of the breakpoint contigs from the secondary assemblies did not span the sequence around the breakpoint in one contig, hence inadequate for studying the break-point region. On the other hand, hifiasm arguably excelled at pre-serving the contiguity of all haplotypes at a phasing stage. Hence, we decided to use hifiasm assemblies for our further analyses.

## Construction of an optical genome map

The CS10 sample from the Celtic and BS3 sample from the Baltic Sea were used for optical (BioNano) mapping[64]. Two mg of frozen testis tissue for each sample was fixed and treated according to the manu-facturer's soft tissue protocol (Bionano Genomics, San Diego, US), except that following homogenization and before fixation, the tissue suspension was passed through a 100 μm cell strainer (Miltenyi Biotec, Gaithersburg, MD). Fixed tissue was washed, and then approximately 0.7 mg was embedded in each of three agarose plugs. Embedded tissue was digested with proteinase K, treated with RNase, washed, and then

equilibrated in Tris-EDTA (TE), pH 8.0. High molecular weight DNA was recovered by digesting the plugs with agarase and cleaned by a dialysis step. DNA was quantified in triplicate by Qubit (ThermoFisher) and diluted with buffer EB (Qiagen) as needed to lower the concentration to <125 ng/μL. DNA was then labeled with the DLS Labeling Kit (Bio-nano Genomics, San Diego, US). Recovery of labeled DNA was verified by Qubit HS dsDNA assay. Labeled molecules were linearized and imaged with the Saphyr® system (Saphyr chip G2.3) to create the molecules data file. The single molecule image data was de novo assembled into optical genome maps using the hybridScaffold pipe-line (Bionano Solve 3.7) with default settings (Supplementary Table 8). The assemblies were visualized using Bionano Access 1.7 webserver.

## Genome alignments of HiFi reads onto reference and PacBio assemblies

The previously reported chromosome level genome assembly[40] was used as a reference to align PacBio HiFi reads using minimap2 (v2.22-r1101)[65]. Alignments with a mapping quality greater than 20 were kept using SAMtools[66] and used for further analyses. Genome-to-genome alignments were carried out using MUMmer (v4.0.0rc1)[67,68] with parameters "nucmer --maxmatch -c 500 -l 200", where all PacBio assemblies were aligned to the reference genome and to each other. The alignments for the inversion regions were visualized as dot plots using the mummerplot function of MUMmer.

## Finding inversion breakpoints using HiFi reads and constructing inversion scaffolds

In our previous study, we used PacBio continuous long reads data from one Celtic Sea individual (CS2) to find the breakpoints for inversions on chromosomes 6 and 17[31], where we visualized the alignment of a single read spanning the breakpoint using IGV[69] and Ribbon[70]. Here, we used the same method to find the breakpoints on chromosomes 12 and 23 using PacBio HiFi reads and verified previously deduced break-points for chromosomes 6 and 17 using PacBio HiFi reads.

To compare inversion haplotypes at the sequence level, it is essential to use inversion regions in the scaffolded form. Although PacBio contigs were highly contiguous, they were not long enough to span the entire inversion regions (ranging from 1.5 to 8 Mb). Hence, we used optical mapping data for scaffolding PacBio contigs. However, the resulting hybrid scaffolds had many gaps and were not contiguous for the entire inversion regions (Supplementary Table 8). To overcome this, we manually curated the Bionano hybrid assemblies in the inversion regions by replacing gaps with PacBio contigs and joining hybrid scaffolds whenever necessary. We followed the NCBI recom-mendation to maintain a gap size of 100 (https://www.ncbi.nlm.nih.gov/genbank/wgsfaq/#q6). The correct order and orientation for PacBio contigs were decided based on their alignment to the reference assembly. The Bionano assemblies used for constructing inversion scaffolds were selected based on the contiguity of hybrid scaffolds for the respective inversion regions. As a result, we used CS10_hap1 and BS3_hap1 assemblies to make inversion scaffolds for Chr6; and CS10_hap1 and BS3_hap2 to make inversion scaffolds of Chr12, 17, and 23. This way, we had one inversion scaffold for each inversion allele for all four inversions. These scaffolds were then used for two purposes—(1) to investigate the structural variants (SVs) in the breakpoint region, and (2) as a reference to scaffold the inversion regions of the remaining 22 PacBio genomes using RagTag (v2.0.1)[71]. The PacBio contigs were selected based on their alignment to the reference genome. The threshold for an alignment block was kept at 10 kb to avoid the incorporation of non-specific contigs. However, some of the non-specific contigs had alignment blocks larger than 10 kb and had to be removed manually. The inversion scaffolds adjusted in this manner were CS7_hap1, BS2_hap1, BS4_hap2, and BS5_hap2 for the Chr17 inversion and CS4_hap2, CS5_hap2, CS7_hap1, CS10_hap2, BS5_hap1 for the Chr23 inversion. To use these scaffolds for further analysis, it was

necessary to have the breakpoints of these scaffolds. However, it was challenging to apply the previously described visualization method using a single read for each scaffolded inversion because of the complexity of the breakpoint region and the presence of multiple SVs in the vicinity of the breakpoints. Hence, we used nucmer in MUMmer[68] to align N and S alleles from CS10_hap1 and BS3_hap2 inversion scaffolds, respectively. The resulting delta files were converted to a paf format using "delta2paf" script from paftools.js in minimap2[65] to obtain the alignment coordinates. As only homologous sequences will align in MUMmer, the coordinates where the alignment changes its orientation would be the breakpoint. We opted for a conservative approach where SVs such as duplications, insertions, deletions, and repetitive sequences at the breakpoint regions were placed outside the inversion. This way, we first obtained breakpoints on CS10_hap1 and BS3_hap2 inversion scaffolds. They were used as a reference to obtain breakpoints from the rest of the inversion scaffolds by finding sequence homology for the 10 kb sequence near the breakpoint using BLAST (v.2.11.0+)[72].

### PacBio assemblies as references for N and S alleles

Although the reference genome assembly is of high quality and contiguous, it is not representative of all SVs and repeat content near the inversion breakpoints because of variation among haplotypes. Hence, we leveraged the accuracy and contiguity of HiFi assemblies and scaffolding of Bionano optical maps to build hybrid scaffolds of two assemblies (CS10_hap1 and BS3_hap2, representative of assemblies with N and S inversion alleles). We used one of each Celtic and Baltic HiFi assemblies as a reference to study structural variations, and repetitive sequences near the inversion breakpoints. As we used CS10_hap1 and BS3_hap2 assemblies to construct most of the inversion scaffolds (7 out of 8), we decided to use the same assemblies as references for structural analysis. The contigs used to build the inversion scaffolds were replaced by the inversion scaffolds in CS10_hap1 and BS3_hap2 genome assemblies. In the case of the Chr6 inversion, the original inversion scaffold was built using BS3_hap1 contigs. However, the length of BS3_hap1 inversion scaffold was the same as that of BS3_hap2 (Supplementary Fig. 15), and hence, no additional modification was done for Chr6.

### Deduction of ancestral inversion allele using European sprat as an outgroup species

European sprat, an outgroup species that diverged from the Atlantic herring 11–12 MYA[37,38] was used to determine the ancestral inversion alleles. We aligned the European sprat reference genome (fSprSpr1.1, GCA_963457725.1) to the Atlantic herring genome using Chromosemble from satsuma2 (v.2016-12-07)[73] and minimap v2.26 implemented in D-GENIES[74], and using MUMmer[68]. With the resulting outputs, we studied the synteny between Atlantic herring and European sprat genomes and chromosomes/scaffolds harboring inversions using circlize in R[75] and dot plots. Further, we aligned the European sprat sequence homologous to N and S inversion alleles using MUMmer[68] and visualized the alignment on the dot plot. The linear orientation of the alignment before and after the breakpoint was used to determine if the S or N allele is ancestral or derived.

### Analysis of structural variants near inversion breakpoints

To study SVs near the inversion breakpoints at the sequence level, we used one-dimensional pangenome graphs and dot plots from the sequence alignments of all inversion scaffolds and a reference sequence (total of 25 sequences for each inversion). For the pangenome graph approach, we used pggb (v0.3.1)[76] to construct graphs and odgi (v0.7.3)[77] to prune the resulting graphs. To ensure that the alignments were of high quality, we tested multiple combinations of -s (segment length) and -p (percent identity) parameters in the mapping step of pggb. We used higher -s value (20,000–50,000) to ensure that the graph structure represents long collinear regions of the input

sequences. We used lower -p values (90–95) because inversion regions including breakpoint regions are more divergent than the rest of the genome. Exact parameters to build and visualize pangenome graphs are found on the Zenodo repository for this paper[78].

### Short-read dataset, alignment, and variant calling

The same 12 samples from Celtic and Baltic Sea used for long-read sequencing were also sequenced on Illumina HiSeq2000 sequencer to generate 2 × 150 bp paired-end reads of nearly 30x coverage. We assessed the read quality using FastQC 0.11.9[79]. We mapped reads to the reference herring genome Ch_v2.0.2[41] using BWA-MEM v.0.7.17[80] sorted reads with SAMtools v1.12[66] and marked duplicates with Picard v2.10.3 (http://broadinstitute.github.io/picard/). To perform genotype calling for each sample, we first used Haplotyper within the Sentieon wrapper (release 201911)[81], which implements GATK4 HaplotypeCaller[82]. We then combined these 12 samples with previously generated high coverage re-sequencing data for 49 Atlantic herring from Baltic Sea, Celtic Sea, North Sea, Norway, Ireland and United Kingdom, and Canada and 30 Pacific herring (Clupea pallasii, the sister species) distributed from the North Pacific Ocean to Norway[31,33]. For these 91 samples, we performed joint calling of variant and invariant sites using the Genotyper algorithm within Sentieon, which implements GATK's GenotypeGVCFs. We removed indels, and filtered genotypes with RMSMappingQuality lower than 40.0, MQRankSum lower than −12.5, ReadPosRankSum lower than −8.0, QualByDepth lower than 2.0, FisherStrand higher than 60.0 and StrandOddsRatio lower than 3.0. Additionally, we also filtered variants that had genotype quality below 20, depth below 2 or higher than three times the average coverage of the individual. These filtered vcf files were the basis for analyses therein.

### Generating consensus sequences for herring and sprat

Consensus genome sequences for each individual were generated for phylogenetic analyses. We used a custom script do_bed.awk[83] to create a bed file with the coordinates of called positions (variant and invariant sites) for each individual from the vcf files, and bedtools complement (v2.29.2)[84] to produce a bed file of non-called positions. We then used samtools faidx (v.1.12) and bcftools consensus (v.1.12) to introduce individual variant and invariant genotypes into the Atlantic herring reference genome and bedtools maskfasta to hard-mask non-called positions in consensus fasta sequences.

Fasta and vcf reference sequences for the outgroup species, the European sprat, were generated by first using Chromosemble from satsuma2 (v.2016-12-07)[73] to align to the sprat assembly generated in this work (see section "Long-read dataset and construction of PacBio genome assemblies") to the Atlantic herring reference genome. Then, using a custom R script ancestral_state_from_sprat.R[85] that uses packages Biostrings (v.2.68.1), biomaRt (v.2.56.1), GenomicRanges (v1.52.0) and tidyverse (v.2.0.0), we extracted the regions of the sprat assembly that aligned to herring genes, choosing the longest sequence if multiple regions aligned to the same gene and excluding sprat sequences that aligned to less than 25% of the total length of genes. Then, we realigned herring and sprat sequences using MAFFT (v7.407)[86]. To keep high-quality alignments of true homologous regions, we further excluded alignments with missing data higher than 20% and proportion of variable sites higher than 0.2, as calculated by AMAS summary[87], resulting in 15,471 alignments. We converted the alignments in fasta format to a vcf file using a custom script ancestral_vcf.py, genoToVcf.py (downloaded in October 2021 from https://github.com/simonhmartin/genomics_general) and bcftools[85]. From the final vcf file, we used the same procedure as above to generate a consensus genome sequence for the sprat in the genomic coordinates of the Atlantic herring.

### Phylogenetic inference

To obtain a maximum likelihood tree for the entire genome and for each inversion, we concatenated individual consensus genome-wide

sequences of all herring individuals and the sprat. We extracted inversion alignments using samtools faidx. We removed all positions with missing data from the whole-genome alignment using AMAS trim, whereas for the inversions, we allowed sites with missing data for at most 50% of the individuals. As this alignment with the sprat contained information only for 15,471 genes (114 Mb alignment, 14% of the genome), we repeated tree inference with alignments containing only herring individuals to retain more positions (346 Mb alignment, 43% of the genome) and improve the inference of intraspecific relationships, rooting the trees on the branch splitting Atlantic and Pacific herring, the typical position of sprat (Supplementary Fig. 8)[37,38]. Maximum likelihood trees were generated with IQ-TREE (v.2.0-rc2)[88] with model selection[89] and 100 ultrafast bootstrap replicates[90]. Phylogenetic trees were visualized with FigTree v.1.4.4 (https://github.com/rambaut/figtree).

## Population genomic analyses

To calculate summary statistics (differentiation as $F_{ST}$, divergence as $d_{xy}$, nucleotide diversity as $\pi$, and linkage disequilibrium as $R^2$) within and between inverted haplotypes, we first determined the genotype of each individual in the dataset using two approaches. First, we used a set of previously ascertained highly differentiated SNPs between the $N$ and $S$ haplotypes at each inversion[31,41] and extracted genotypes for all individuals at those positions using bcftools view, keeping only positions that were polymorphic and biallelic in our dataset. The final plots were produced using the R packages tidyverse, ggplot2 (3.4.2) and ggrstar (1.0.1). Second, we performed a principal component analysis (PCA) using biallelic SNPs with less than 20% missing data and minor allele frequency (maf) above 0.01 and all individuals from the Baltic and Celtic Sea in our dataset ($n = 35$) across sliding windows of 200 SNPs for chromosomes 6, 12, 17 and 23 using lostruct (downloaded October 2022 from https://github.com/petrelharp/local_pca)[91]. We genotyped individuals by plotting the first principal component for each individual across inversion regions using ggplot2[92].

The genotype information obtained by these methods was further used to make four groups namely (1) all Atlantic herring ($n = 61$), (2) all Pacific herring ($n = 30$), (3) all Baltic Sea herring homozygous for $N$ alleles ($n_{chr6} = 20$, $n_{chr12} = 19$, $n_{chr17} = 16$, $n_{chr23} = 16$), and (4) all Celtic Sea herring homozygous for $S$ alleles ($n_{chr6} = 15$; $n_{chr12} = 15$, $n_{chr17} = 15$, $n_{chr23} = 13$; Supplementary Table 9). Using a vcf file containing variant and invariant sites, we selected sites with less than 20% missing data and maf > 0.01, we calculated $d_{xy}$ and $F_{ST}$ between these groups and $\pi$ within groups in 20 kb sliding windows using pixy (v.1.2.5)[93]. We also estimated pairwise $F_{ST}$ between populations excluding inversion regions using the same approach, and calculated the population migration rate as $[(1/F_{ST}) - 1]/4$ (ref. [53]). To study patterns of recombination suppression caused by the inversion, we calculated $R^2$ in vcftools for both groups of homozygotes combined or individually (v.0.1.16)[94]. For computational reasons, we used a more conservative filtering; we kept sites with less than 10% missing data, genotype quality above 30, maf above 0.1 and thinned SNPs so that they were at least within 5 kb of each other.

## Estimating the age of the inversion

We used $d_{xy}$ between Atlantic and Pacific herring individuals and between $N$ and $S$ homozygotes for each inversion to calculate the net nucleotide diversity as $d_a = d_{xy} - (d_x + d_y)/2$[95]. $d_a$ was then used to estimate divergence time between Atlantic and Pacific herring, and between $N$ and $S$ inversion haplotypes. Assuming a mutation rate per year of $\lambda = 3.3 \times 10^{-10}$ (ref. [50]), we use the formula $T = d_a/2\lambda$ to calculate the divergence time between Atlantic and Pacific herring, and between the $N$ and $S$ haplotypes[95].

## Mutation load

To understand if recombination suppression between inverted haplotypes had resulted in the differential accumulation of deleterious

mutations in inversion haplotypes, we took three main approaches. We used a similar sampling for all the analyses described above, grouping individual homozygotes for the $N$ or $S$ allele at each inversion to study haplotype differences. In all analyses, we used European sprat as the outgroup.

First, we calculated the ratio of number of substitutions per non-synonymous site ($d_N$) to the number of substitutions per synonymous site ($d_S$), or $d_N/d_S$ between the $N$ or the $S$ haplotype and European sprat for all genes inside the inversions and for all genes in the genome using all 61 Atlantic herring individuals. We extracted the coding sequence of the longest isoform for each gene for each homozygote, using the consensus genome-wide sequences for each individual as described before and a combination of agat (v.0.8.0)[96], and bedtools getfasta. Then, using $dnds.py$[85] that implements biopython (v.1.79, https://biopython.org/), we calculated a consensus sequence for the $N$ and $S$ haplotypes using the sequences of homozygotes, converting any ambiguous positions or stop codons into missing data and removing gaps from the alignment. Finally, using alignments longer than 100 codons, we used the $cal\_dn\_ds$ function from biopython to calculate $d_N/d_S$ using the M0 model from codeml[97]. We finally excluded alignments where values of $d_N/d_S$ were higher than 3, assuming that these could be caused by alignment issues between the herring and sprat genomes. We plotted values using ggplot2 and performed a two-sided $t$-test in R 4.3.0 to test for significant differences in $d_N/d_S$ between $N$ and $S$ haplotypes, and between haplotypes and the genome-wide $d_N/d_S$ distribution.

Second, we compared the site frequency spectrum of derived non-synonymous and synonymous mutations between $N$ and $S$ inversion haplotypes, using SNPEff (v5.1)[98] to classify the functional impact of SNPs segregating among all 61 Atlantic herring individuals. We then used a combination of bcftools and vcftools, to calculate the frequency of derived non-synonymous and synonymous biallelic SNPs (vcftools options --freq and --derived), in sites with less than 20% missing data. To run this analysis in vcftools, we used bcftools to add an extra field called $AA$ to vcf files with the European sprat genotype to be used as outgroup when calculating derived allele frequencies.

Finally, we compared the proportion of transposable elements (TE) between $N$ and $S$ haplotypes. In this case, we used the assembled genomes for Baltic and Celtic Sea individuals and the Repeat-Masker pipeline to annotate TEs. We first used RepeatModeler (2.0.1)[99], and the hap1 assemblies of individuals CS4, CS7, BS3 and BS4, which were either the longest and/or more contiguous of each CS and BS assemblies (Supplementary Table 1), to identify and model novel TEs. We also used BLAST (v.2.11.0+)[100], to compare all TEs to all protein-coding herring genes and filtered out TEs that mapped to genes. To improve the final annotation of the TEs in our database, we compared unknown repeat elements detected by RepeatModeler with transposase database (Tpases080212)[101] using BLAST and used TEclassTest (v.2.1.3c)[102] to improve the classification of TEs in our database. We combined all four databases into one and removed redundancy with CDHit (v4.8.1)[103] with the parameters -c 0.9 -n 8 -d 0 -M 1600. Then, we used this library as input for RepeatMasker[99] to annotate and mask TEs in all the assemblies. We also annotated TEs for the inversion region of each assembly. To determine the coordinates of each inversion for each individual assembly, we used the scaffolded inversions for CS10 and BS3 (described above). We extracted 10 kb regions immediately after and before the breakpoints of the inversions from CS10 and BS3 and mapped them to the other individual assemblies using BLAST and detected the breakpoints of the inversions in each assembly with a custom script $find\_breakpoints.py$ that parsed the BLAST output[85].

## Screening for regions of recombination within inversions and enrichment of genetic variants

To further study the occurrence of gene flux between $N$ and $S$ inversion haplotypes, we inspected the allele frequency differences

of minor alleles in *NN* and *SS* individuals combined. We first determined which allele (reference or alternative) was the minor allele in a vcf file combining *NN* and *SS* homozygotes ($n$ = 35). We removed sites with minor allele frequencies below 0.2 in this combined vcf, to remove invariant sites that create noise in our dataset. Then, we (1) calculated the frequency of the minor allele in *NN* and *SS* groups separately, (2) determined the absolute difference between these frequencies or dAF, and (3) divided dAF by the maximum frequency of the minor allele (MAFmax) in *NN* or *SS*, obtaining what we call delta allele frequency prime (dAF'). The rationale is that a new mutation occurring in *N* or *S* haplotypes will be in low frequency and will not be shared between haplotypes. As an example, consider a SNP with freq(*N*) = 0.2 and freq(*S*) = 0. dAF and dAF' for this variant will be 0.2 and 1.0, respectively, since dAF' = abs(0.2-0.0)/0.2 = 1. If gene flux has occurred, however, the combined minor allele can be shared between inversions, resulting in lower values of dAF' (e.g., freq(*N*) = 0.2 and freq(*S*) = 0.3, resulting in dAF' = 0.33). In a situation where no gene flux has occurred between inversion alleles dAF' would be 1.0 for all SNPs, and the results for the Atlantic herring inversions reveal major deviations from this prediction (Fig. 8).

We inspected the function of variants in different dAF categories, by performing an enrichment analysis. First, we used SnpEff (v3.4)[98] to annotate the genome-wide variants and classify them into various categories (non-synonymous, synonymous, intronic, intergenic, 5'UTR, 3'UTR, 5 kb upstream, 5 kb downstream). Only sites within the inversion were kept for further analysis. The expected number of SNPs in each category for each inversion was calculated as p(category) X sum(extreme), where p is the total proportion of a specific SNP category without any dAF filter and sum(extreme) is the total number of SNPs with dAF > 0.95. A standard $\chi^2$ test was performed to test the statistical significance of the deviations of the observed values from expectation. We particularly looked at the genes that are extremely differentiated (dAF > 0.95) in the non-synonymous category.

### Reporting summary

Further information on research design is available in the Nature Portfolio Reporting Summary linked to this article.

## Data availability

The long-read and short-read sequence data generated in this study have been submitted to the NCBI database under accession number PRJNA1023520. De novo genome assemblies constructed using PacBio data have been deposited to the NCBI database under accession number PRJNA1158307 for CS2_hap1.fa, PRJNA1158306 for CS2_hap2.fa, PRJNA1158305 for CS4_hap1.fa, PRJNA1158304 for CS4_hap2.fa, PRJNA1158303 for CS5_hap1.fa, PRJNA1158302 for CS5_hap2.fa, PRJNA1158301 for CS7_hap1.fa, PRJNA1158300 for CS7_hap2.fa, PRJNA1158299 for CS8_hap1.fa, PRJNA1158298 for CS8_hap2.fa, PRJNA1158297 for CS10_hap1.fa, PRJNA1158296 for CS10_hap2.fa, PRJNA1158295 for F1_hap1.fa, PRJNA1158294 for F1_hap2.fa, PRJNA1158293 for F2_hap1.fa, PRJNA1158292 for F2_hap2.fa, PRJNA1158291 for F3_hap1.fa, PRJNA1158289 for F4_hap1.fa, PRJNA1158288 for F4_hap2.fa, PRJNA1158287 for F5_hap1.fa, PRJNA1158286 for F5_hap2.fa, PRJNA1158285 for F6_hap1.fa, PRJNA1158284 for F6_hap2.fa, PRJNA1158283 for NSSH2_hap1.fa, PRJNA1158282 for NSSH2_hap2.fa, PRJNA1158281 for NSSH10_hap1.fa, and PRJNA1158280 for NSSH10_hap2.fa. The European sprat PacBio reads have been submitted to the NCBI database under accession number PRJNA1023385. European sprat de novo genome assembly generated in this study and the inversion scaffolds have been deposited to FigShare (https://doi.org/10.6084/m9.figshare.24354943). This study also used previously available short-read whole-genome sequencing data from accession number PRJNA642736 in the NCBI database.

## Code availability

The analyses of data have been carried out with publicly available software and all are cited in the "Methods" section. Custom scripts used are available in https://zenodo.org/records/12786351 and https://zenodo.org/records/12792460.

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

## Acknowledgements

We are grateful to the scientists and crew on the Marine Institute's Irish Groundfish Survey for collecting herring samples from the Celtic Sea. The study was supported by the Knut and Alice Wallenberg Foundation (KAW 2023.0160) and Vetenskapsrådet (2017-02907). Computational infrastructure was provided by the National Academic Infrastructure for Supercomputing in Sweden, partially funded by the Swedish Research Council through grant agreement no. 2022-06725. M.S.F. was funded by an MSCA European Postdoctoral Fellowship (Project 101063864, INVERT2ADAPT) granted by the European Research Executive Agency.

## Author contributions

L.A. conceived the study. M.J. and M.S.F. performed all bioinformatic analyses. E.F. contributed to sample collection. M.E.P. and B.W.D. contributed to the bioinformatic analysis. M.J., M.S.F., and L.A. wrote the paper with input from other authors. All authors approved the paper before submission.

## Funding

## Competing interests

The authors declare no competing interests.
