## [Peer Review File · Nature Communications]

The origin and maintenance of supergenes contributing to ecological adaptation in Atlantic herringREVIEWER COMMENTS

Reviewer #1 (Remarks to the Author):

Dear Jamsandekar et al.,

I have now read through the submitted ms entitled “The origin and maintenance of supergenes contributing to ecological adaptation in Atlantic» and have to say that I really enjoyed the ms. It is a piece of solid work that they have conducted. I appreciate the way they have taken advantage of long read sequencing to in depth characterize the inversion break points as well as their evolutionary history. However, before any re-submission I do have some major concerns and questions that I think should be looked into.

First, I do think that they should re-think the order of some of their results. I would strongly suggest that you move the “Structural variations at the breakpoint regions” up so that it comes after or as part of the very first section of the results “Characterization of inversion breakpoints on chromosomes 6, 12, 17, and 23”. I think these two paragraphs belongs together. Further, some of the statements in the first paragraph is not consistent with the statements in the other. So, I also think you need to modify your writing. For instance, you write on line 123-125: “We found similar breakpoint co-ordinates for all samples in each inversion (Fig. 1, Supplementary Table 2), suggesting that these inversions have originated just once, stemming from a one-time break in the chromosome,”

What you write is not precisely what you see. I think you should state “near similar” instead of “similar” – since this is more the case – and I would also suggest that you define how similar they are: i.e. the range of bp difference detected (to be included in the Supp Table 2). In this way the reader can easily see how similar they are – even if not completely the same (and agree with the authors that based on their findings the inversions seem to have a single origin!). And would in fact say that this table (Supp Table 2) should be presented as the first main table in the ms, since this table display their findings in a good way: showing how similar they are between the different individuals inspected.

But then one question: Why only four and not six individuals for the break points detected on Chr 6? Not stated anywhere in the ms why two of the scorings here were not listed. And why only approximate breakpoints for 12? This is also not explained anywhere (as I can see at least). If I understand correctly, it is Chr 17 and 23 that have iffy breakpoints not Chr 12?

And most importantly the referring to Figure 1 in this regard is wrong – this does not show the near similar breakpoints. Think you should move the ref to Figure 1 to the lines above (119-120), since Figure 1 do show a schematic overview of the identification of the breakpoints. Here, I would say that the authors should really re-think how they present this overview. Figure 1 is a very complex figure and not easy to assess. I acknowledge that what you try to show is complex, but I think that with some edits this figure would be more understandable for the reader. For instance, instead of how it is now illustrated you could show how the PacBio read is split into two parts and how the orientation is shifted on the PacBio read itself and not on the reference (this is misleading I think). This can be shown below the actual reference with arrows (showing the PacBio read), and then from these arrows (showing the split) show the alignments onto the full PacBio read below here again ... think this could help on the readability of the figure
And the on top only have the candidate genes listed as before.

Further you state: “We re-evaluated the breakpoints obtained in our previous study for Chr6

and Chr17 inversions and found 1-3 kb shifts at two positions (Supplementary Table 2).”

These shifts are not indicated in the table. Would suggest that you add a second column to show these shifts properly. And maybe state why you see these shifts. I would assume due to genome assembly differences (that the newer assemblies generated within this publication is of higher quality)?

Additionally, I would suggest that the confirmation of the N and S scaffolds generated by optical mapping data could be placed in the supplement (not needed as a main figure I think). Since quite similar to what we see in Figure 4.

Then to the next section: “Identification of ancestral haplotypes using European sprat as an outgroup” which I suggest to move further down in the results section as a part of the “Origin of inversion haplotypes”.

These sections belong more or less together, I think. And I also want to point out that the authors should be a bit more cautious about their interpretations when only using an outgroup species as the main comparison to define ancestral vs derived state of an inversion. They cannot be 100% sure that this outgroup species does not harbour any inversions and/or SV in these genomic regions, and not at least which state of is represented in the genome sequenced.

That said, they do show that Pacific herring do not harbour the same inversion as Atlantic herring, which again support their hypothesis. However, the maximum likelihood trees of the inversion haplotypes and the IGV plots do indicate that also Pacific herring have some SV in these regions that should be looked into. And what is the state of this region (at the population level) in sprat? I would assume nobody has looked into yet? However, if the authors do have additional information here from the mentioned Pettersson M. E. et. al. manuscript that would be of importance for this interpretation – I would encourage the authors to give the reviewers get access to and/or insight into these results so that we can fully evaluate these statements.

And some additional comparisons of the sprat genome vs Atlantic herring genome would have helped. Is the chromosome nr the same in these species? Is the homology/synteny OK between the homologous chromosomes? Based on this I would suggest that Figure 3 could have included additional dot plots of the all chromosomes and not only the inversion break points. This would also aid in the interpretation of the inversion region, and how solid these results are. If no further information is given, I would strongly suggest to tone down the statements in the ms, and define inversions as “putatively” ancestral and/or putatively derived. If Figure 3 should be part of the main ms or moved to the Extended data should be based upon if they are willing to show more of the comparative work here or not, I think.

For the section “Structural variations at the breakpoint regions» that I suggest you to move up – I also have some specific comments:

In line 180-183 you write: “The sequence alignments of N and S alleles near the breakpoints indicated that the breakpoints for all four inversions were flanked by inverted duplications ranging from 8-60 kb in size and contained one or no gene (Fig. 4 and Extended data fig. 4).»

As far as I can see from Figure 4 and other statements later on in the ms (see f. ex line 373-374) – this is not the case for the inversion on Chr 23. So, I would assume that you should correct this sentence!

And PS would also state that I truly love Figure 4! And think that moving this up would also be good for the ms itself (as part of moving this section up!). So, this now becomes Figure 2.

Moreover, I really find your observation about the distal breakpoints being divergent between the individuals inspected as highly intriguing! You write in line 188-190: “For instance, distal breakpoints of all inversions were divergent among individuals, revealing the existence of non-shared structural variants. In particular, the Chr17 and Chr23 breakpoints were the most complex.”

However, these results are now only displayed by Figure 4, which is a comparison between two individuals, i.e. the CS10_hap1 and BS3_hap2 genome assemblies. Could this be plotted for the other individuals too? And shown as a Supp Figure? Additionally, I suggest that you also refer to the Supp Table 5 here (where you list all the different SV in more detail). And one additional question how different are these sequences? Would be good to have some estimations regarding the sequence homology.

Further you state: “The distal breakpoint of Chr17 coincided with a telomeric sequence that varies in length (0-300 kb) outside the breakpoint and that is misaligned in the genome graph.”

This finding should have been visualized on the genome graph on Extended Figure 5 I think – do not catch this finding from the Figure as is. And I would also strongly suggest that for ALL figures that includes sequence/chromosome alignments to include bp position (such as Extended Figure 5 amongst others). On this figure the breakpoint position should also be noted.

For the next section where I have some issues is: “The evolutionary history of inversion haplotypes”. Here, in line 256-261 you state the following:

“The inversion regions showed strong differentiation between N and S homozygotes (high F_{ST}) which is in sharp contrast with the flanking regions (low F_{ST}). An exception to this is a region proximal to the Chr17 inversion breakpoint. A careful inspection of our PacBio data showed that this is not part of the inversion and must be a sequence polymorphism in very strong LD with the inversion polymorphism.”

This careful inspection needs to be shown. What did you find? From the information given and what is displayed in Figure 6, 8 and Extended Data Figure 2 and 5 – this looks suspiciously like a miss-assembly – and that this should be part of the inversion: by flipping/reorient the region outside of the inversion. Especially when looking at the Extended Data Figure 2, both the IGV (first part of the inversion) and the sliding window PCA. Do you have PacBio reads spanning this region to confirm this or not. This needs to be revisited and/or better documentation for not being part of the inversion should be included!

For the IGV plot also here bp position should be denoted. Additionally, I would recommend that they display not only the inversion but spanning some parts outside of the inversions – by doing this it would be easier to see the inversions, and if the inversions are built up by more than one inversion (i.e. being a nested or double inversion). For me, this could in fact be the case for the inversion on Chr 17 and 23. I.e. the beginning of 23 as well as the beginning of 17 having a different pattern, indicating that they are built up by a double/nested inversion?

But here, I would for Chr17 look if this is due to a miss-assembly or not as I suggested above.

PS: I truly like this figure (Extended Data Figure 2) – and would consider moving this one after some modifications to become part of the main ms.

Then for the inversions on Chr 17 and 23 showing a little less pronounced F_{st} and LD

patterns than the inversion on Chr 6 and 12. Can you speculate why you see such differences between the inversions?

Can this difference be due to these ones being of an older age and/or younger age (even if this was not shown by the divergence estimations?). Or could it be due to these ones are built up of double inversions and increased recombination? Seems like you at least show that the linkage is weaker for these inversions.

In line 292-293 you write: "Furthermore, the observed derived alleles at high frequencies are candidate mutations for being under positive selection."

Not sure if this is part of your observations/results or as an intro to the following sentence: "Further, we compared the dN/dS ratio for the N and S alleles at each locus in an attempt to find genes that may show accelerated protein evolution as part of the evolution of these adaptive haplotypes."

Please revisit this part of the text and clarify.

If Figure 7 is needed as part of the main ms can be re-evaluated, I think. Could easily be moved to the Extended Data Section.

For the intro of the discussion: I would say that the authors should try to shorten it a bit but also here in the intro mention that they have done a detailed characterization of the breakpoint regions and thus, enable to define how the originated (by ectopic recombination). This is as I see it an important part of the work conducted and now missing from the intro of the Discussion.

Then, I have one additional question. Did not the optical mapping conducted on CS10 and BS3 generate chromosome level genome assemblies? You do display results of the whole chromosomes of Chr 6, 12, 17 and 23. I guess you also have the whole chromosomes for the other chromosomes? If yes, the fact that you do generate two new high quality reference genomes for Atlantic herring should be highlighted.

This will be a really nice resource that should be released for the research community, I think!

In this regard I also want to point out that in line 530-531 you write:

"However, the resulting hybrid scaffolds had many gaps and were not contiguous for the entire inversion regions (Supplementary Table 1)."

The hybrid genomes are not listed in this table Please provide the data needed to show this.

PS: This is also the case for other results presented I think – that the authors in several occasions refer to tables and figures that do not show/display the results that they refer to. And maybe more importantly several times also describe results that have no table or figure at all. Please go through the entire ms – and present the result that is needed in a proper way.

Finally, and maybe most importantly, the authors need to revisit their definitions of the breakpoints, I think. First, of all not sure why you have defined that the inversions should not harbour the inverted duplications in the distal end for Chr 6, 12, and 17 and for some of the inversions also in the proximal end (for Chr 6 and 12)?

For me this seems rather odd since these genomic regions are inverted and should in my opinion part of the inversion. This is also how it seemingly looks like when looking at Figure 4. If this is not what you see when using the PacBio reads this should be presented in the ms as a figure/table, I think.

In this regard I also want to mention: When looking at some of the figures the definition of the breakpoints seems not to be consistent. I suggest that you have a closer look at this once more.

F. ex for Chr 17 on Figure 6 the light grey marking show that the inversion spans the entire distal end of that Chromosome. This is not as is shown on Figure 4, Figure 2 nor the Extended Data Figure 4. But what you see when you look at the LD in Figure 6 as well as what you see on Figure 8 (the dAF' analyses) and the Supp Figure 2. Please revisit this. And for the Supp Figure 2 I also notice that you here have included the suspicious region (that I mentioned above) as part of the inversion, which I think is correct. But if this is the case, this needs to be looked into and changed in other part of the ms.

PS. For Figure 6 C and D the Chromosome naming is lost (not shown), probably hidden by Figure 6E?

And as a last comment: I again have to point out that I truly enjoyed reading the ms and I really think that the work presented is substantially improving our knowledge on the evolutionary role of larger chromosomal inversions.

Reviewer #2 (Remarks to the Author):

This manuscript presents the long-read assemblies of 12 Atlantic herring individuals and an analysis of population short-read re-sequencing analysis of 49 Atlantic herring and 30 Pacific herring. This study extends the work of Han et al (2020) that identified four major inversions that differentiate northern versus southern populations of Atlantic herring. Han et al (2020) did a short-read sequencing study of 35 to 100 herring and used the data to identify population specific genetic differences. The inversions were identified based on clusters of differentiated SNPs between the populations. Han et al. (2020) suggested that major shifts in allele frequencies at candidate loci within the inversions bear molecular population genetic signatures of selection that are correlated with environmental differences between northern and southern populations such as temperature and salinity.

This study uses long-read read sequencing to assemble genomes that identify inversion breakpoints in the northern (Baltic Sea n=6) and southern (Celtic n=6) Atlantic herring collections. The goals of this study are:

1. to determine the mechanisms that generate inversions by mapping breakpoint coordinates and structural variants (SVs) around the breakpoints;
2. to infer the origin of inversions by determining its ancestral state and age using European sprat (*Sprattus sprattus*) as an outgroup species;
3. to use phylogenetic analysis to infer the evolutionary history of inversions;
4. to examine the effects of suppressed recombination by analyzing patterns of variation, differentiation, linkage disequilibrium, mutation load, and gene flux (genetic exchange between inversion haplotypes).

The major findings of the paper are:

1. The paper describes the generation of 12 high quality long-read assemblies of Atlantic herring. One of the challenges for assembly construction were for individuals heterozygous for the northern and southern inversions. The assembly algorithm hifiasm did create primary and secondary contigs for the two arrangements, however, HiCanu did not infer contiguous sequence through the breakpoints of the secondary arrangements. Optical genomic maps

were used to confirm the assemblies and scaffold contigs across the inverted regions.

2. Breakpoints were mapped for four inversion events on four chromosomes. The derived inversion is of unique origin based on shared breakpoint coordinates for three of the four inversion events. Inverted repeats are found within the breakpoint regions suggesting that pairing of the repeats and ectopic exchange generated the inversions. None of the breakpoints disrupted the coding regions of genes rejecting one aspect of the unreferenced position effect hypothesis for the establishment of inversions (KIRKPATRICK AND BARTON 2006). The position effect hypothesis cannot be completely ruled out because this study does not present transcriptomic data to test whether the breakpoints disrupted gene expression of boundary genes (PUIG et al. 2004).

3. The ancestral and derived arrangements was inferred for the four inversion polymorphisms using the sprat outgroup species. The four inversion polymorphisms are > 1.5 million years old, which is consistent with the comparable levels of nucleotide diversity between ancestral and derived arrangements. Ancestral and derived arrangements are highly differentiated from each other. Consistent with this is the high levels of linkage disequilibrium, but not all sites are in LD with each other, which suggests that gene flux may be breaking up LD as has been seen in *Drosophila* (FULLER et al. 2017).

4. Divergent selection was inferred based on maintenance of alternative haplotypes in the face of high gene flow, however, no data or citations are provided to support the contention that gene flow is sufficient to prevent differentiation.

5. No evidence for genetic load based on dn/ds ratios, Site frequency spectra are similar for synonymous and nonsynonymous sites and the proportion of TEs between N and S inversions is similar.

6. There is evidence for genetic flux among inversion types based on allele sharing among arrangements. There is mixture of sites that show lack of exchange with sites that show extensive sharing. Sharing is uniform across the inverted regions of the four inversion polymorphisms. This pattern is consistent with the LD plots that show low and high LD among sites. This pattern is expected for old inversion polymorphisms.

7. The inversion polymorphisms shows clinal variation associated with different environments.

Strengths. This manuscript presents new long-read sequence assemblies for Atlantic herring that have four inversion polymorphisms on four chromosomes. Studies of inversions until recently have been limited to species with methods that allow detection with direct approaches. Comparative genomics with techniques like long read sequence allow the detection of inversions in non-model organisms. The data presented here allow the analysis of inversion breakpoints between ancestral and derived arrangements providing insights into the mutational mechanism that generates inversions. These data also allow the inference of the ancestral and derived arrangements as well as estimates of the time of origin of the chromosome. Population re-sequencing data provide data on the structure of nucleotide diversity within and between the arrangements.

Weaknesses. The paper could do better job of framing the work as testing hypotheses about the origin, establishment, and maintenance of inversions (KIRKPATRICK AND BARTON 2006). Figure 1 that shows the comparison of ancestral and derived inversions is confusing. The paper would benefit from analysis of gene flow to make the case for haplotype differentiation in the face of gene flow.

Overall assessment: This paper addresses important questions in evolutionary biology, i.e., how do karyotypic variants arise in populations and how are they established and maintained. Until genomic approaches were developed, the number of systems where

chromosomal evolution could be studied were limited. With the ability to sequence complete genomes now allows chromosomal evolution to be studied in diverse organisms such as the Atlantic herring in this study. The overall analyses are sound and are reproducible based on the described methods and the conclusions follow from the analyses.

Comments

1. Lines 36-37. Whether alleles within an inversion are inherited as a single unit needs to be tested. There is evidence from *Drosophila* that homologous gene conversion can transfer alleles between different inversion backgrounds despite reduced levels of crossing over.
2. Lines 139-140 Figure 1. This figure is confusing. For example in A, the chromosome 6 inversion is 2.5 Mb long, yet a single 25 kb PacBio read is shown mapped to breakpoints on both ends of the inversion. I assume that the reference genome is one inversion type and the PacBio Read comes from the alternative inversion type, but this is not clear from the Figure legend. An additional issue is why is only one breakpoint PacBio read shown for each inversion event. It would also help to show the inferred coding genes on the PacBio read. Calvete et al (2012) showed that gene duplications and deletions of genes can accompany inversion events in *Drosophila*. This may not necessarily be the case, but the paper should indicate whether gene structure at the breakpoint boundaries is conserved or not. The dot plots in the boxes in B and C are hard to see.
3. Lines 160-172 and 577-584. Inferring the ancestral state of the inversions. This section could be shortened by skipping the discussion of using levels of nucleotide diversity to infer ancestral versus derived state. The nucleotide diversity analysis may have been done to infer the derived arrangement, but because it did not provide a definitive answer, the text from Line 161 to 167 can be deleted. A more direct method is to use the configuration of genes adjacent to the breakpoints as markers of the inversion event using *Sprattus sprattus* to polarize the mutation event, which was done here. If *Sprattus sprattus* has a/b and c/d at the proximal and distal breakpoints, then the derived inversion will likely have the configuration a/c and b/d at the proximal and distal breakpoints (See Supplemental Figure S4 showing the conversion of the Standard to Arrowhead arrangement for an example. FULLER et al. 2017).
4. Line 198 and Line 390. Are the repeats at the breakpoints of the different inversions similar to each other suggesting a common mutational motif?
5. Line 261-262 and Figure 6. Was linkage disequilibrium tested between all pairs of SNPs?
6. Lines 310-312. Are the site frequency data presented for just the genes within the inverted regions? If so, it would be useful to contrast these data with collinear regions. If the SFS are similar, then this would support the conclusion that inverted regions are not accumulating deleterious alleles, i.e., no genetic load.
7. Lines 281-297 It would be useful to know what the frequencies of the three inversion genotypes are for each inversion polymorphism in the different regions. Are they in HWE? If there are deleterious alleles present, one might expect to observe a deficiency of homozygotes and overabundance of heterozygotes.
8. Lines 403-408. The U-shaped pattern of divergence is expected for neutral inversion polymorphisms (NAVARRO et al. 2000). A non-U-shaped pattern might reflect selected genes within the central part of the inverted region that are responsible for the establishment and maintenance of the different arrangements (GUERRERO et al. 2012). In addition to the *D. subobscura* example, consider Figure 7 in Fuller et al. (2017) as example of the maintenance of differentiation in central inverted regions.
9. Lines 414-419. The maintenance of differentiation in the face of extensive gene flux supports the idea that selection is removing transferred genes that create maladaptive phenotypes.
10. Lines 439-443. What is the evidence for gene flow? The data presented here could be

used to estimate the migration parameter Nm , where N is the effective population size and m is the migration rate. One would want to use data from non-inverted chromosomes to estimate Nm . If Nm is greater than 1, than migration is sufficient to homogenize gene frequencies genome-wide. Genes or inversions that are differentiated in the face of this level of gene flow are likely to be selected (HOEKSTRA et al. 2005).

11. Lines 730-744. Betran et al. (1997) developed a method to infer genetic flux in the form of gene conversion and is implemented in DNASP (ROZAS et al. 2017). The method infers conversion tracts using similar reasoning as described here, but allows one to estimate the gene conversion parameter.

Editorial Comments

1. Line 42. Change “single unit” to “single copy” The paper is referring to the unique occurrence of a new inversion mutation where the chromosome breaks at two sites and is rejoined in reverse orientation.

2. Lines 114-115 Table 1 and Supplemental Table 1A. There should be a reference to BUSCO for the completeness analysis (SIMÃO et al. 2015).

3. Line 372. A citation to Richards et al. (2005) could be added here for another example from *Drosophila*.

4. Lines 1013-1014 Table 1 is duplicated.

Literature Cited

Betran, E., J. Rozas, A. Navarro and A. Barbadilla, 1997 The estimation of the number and the length distribution of gene conversion tracts from population DNA sequence data. *Genetics* 146: 89-99.

Calvete, O., J. Gonzalez, E. Betran and A. Ruiz, 2012 Segmental duplication, microinversion, and gene loss associated with a complex inversion breakpoint region in *Drosophila*. *Mol Biol Evol* 29: 1875-1889.

Fuller, Z. L., G. D. Haynes, S. Richards and S. W. Schaeffer, 2017 Genomics of natural populations: Evolutionary forces that establish and maintain gene arrangements in *Drosophila pseudoobscura*. *Molecular Ecology* 26: 6539-6562.

Guerrero, R. F., F. Rousset and M. Kirkpatrick, 2012 Coalescent patterns for chromosomal inversions in divergent populations. *Philosophical Transactions of the Royal Society B: Biological Sciences* 367: 430-438.

Han, F., M. Jamsandekar, M. E. Pettersson, L. Su, A. P. Fuentes-Pardo et al., 2020 Ecological adaptation in Atlantic herring is associated with large shifts in allele frequencies at hundreds of loci. *Elife* 9.

Hoekstra, H. E., J. G. Krenz and M. W. Nachman, 2005 Local adaptation in the rock pocket mouse (*Chaetodipus intermedius*): natural selection and phylogenetic history of populations. *Heredity* 94: 217-228.

Kirkpatrick, M., and N. Barton, 2006 Chromosome inversions, local adaptation and speciation. *Genetics* 173: 419-434.

Navarro, A., A. Barbadilla and A. Ruiz, 2000 Effect of inversion polymorphism on the neutral nucleotide variability of linked chromosomal regions in *Drosophila*. *Genetics* 155: 685-698.

Puig, M., M. Caceres and A. Ruiz, 2004 Silencing of a gene adjacent to the breakpoint of a widespread *Drosophila* inversion by a transposon-induced antisense RNA. *Proc Natl Acad Sci U S A* 101: 9013-9018.

Richards, S., Y. Liu, B. R. Bettencourt, P. Hradecky, S. Letovsky et al., 2005 Comparative genome sequencing of *Drosophila pseudoobscura*: Chromosomal, gene and cis-element evolution. *Genome Research* 15: 1-18.

Rozas, J., A. Ferrer-Mata, J. C. Sánchez-DelBarrio, S. Guirao-Rico, P. Librado et al., 2017 DnaSP 6: DNA Sequence Polymorphism Analysis of Large Data Sets. *Molecular Biology*

and Evolution 34: 3299-3302.

Simão, F. A., R. M. Waterhouse, P. Ioannidis, E. V. Kriventseva and E. M. Zdobnov, 2015
BUSCO: assessing genome assembly and annotation completeness with single-copy
orthologs. Bioinformatics 31: 3210-3212.

Response to the Reviewer's comments

We are grateful for the high-quality reviews we got on this paper. The Reviewers have done an excellent job to highlight strengths in our paper and parts that need to be improved. This has helped us to improve the paper and we are grateful for that. Our point-by-point responses to these comments are given in bold below. The references to specific lines in the manuscript refer to the revised version of the manuscript.

REVIEWER COMMENTS

Reviewer #1 (Remarks to the Author):

Dear Jamsandekar et al.,

I have now read through the submitted ms entitled “The origin and maintenance of supergenes contributing to ecological adaptation in Atlantic» and have to say that I really enjoyed the ms. It is a piece of solid work that they have conducted. I appreciate the way they have taken advantage of long read sequencing to in depth characterize the inversion break points as well as their evolutionary history. However, before any re-submission I do have some major concerns and questions that I think should be looked into.

First, I do think that they should re-think the order of some of their results. I would strongly suggest that you move the “Structural variations at the breakpoint regions” up so that it comes after or as part of the very first section of the results “Characterization of inversion breakpoints on chromosomes 6, 12, 17, and 23”. I think these two paragraphs belongs together.

>>>We have made this change.

Further, some of the statements in the first paragraph is not consistent with the statements in the other. So, I also think you need to modify your writing. For instance, you write on line 123-125: “We found similar breakpoint co-ordinates for all samples in each inversion (Fig. 1, Supplementary Table 2), suggesting that these inversions have originated just once, stemming from a one-time break in the chromosome,”

What you write is not precisely what you see. I think you should state “near similar” instead of “similar” – since this is more the case – and I would also suggest that you define how similar they are: i.e. the range of bp difference detected (to be included in the Supp Table 2). In this way the reader can easily see how similar they are – even if not completely the same (and agree with the authors that based on their findings the inversions seem to have a single origin!). And would in fact say that this table (Supp Table 2) should be presented as the first main table in the ms, since this table display their findings in a good way: showing how similar they are between the different individuals inspected.

>>>We have changed this sentence to “similar, but not identical,” (line 149-150). We have also followed the advice to present Supp Table 2 as Table 2. In this table we have added a column “Shift from consensus breakpoint in bp (proximal/distal)” to indicate the difference in breakpoint co-ordinates among samples.

But then one question: Why only four and not six individuals for the break points detected on Chr 6? Not stated anywhere in the ms why two of the scorings here were not listed. And why only approximate breakpoints for 12? This is also not explained anywhere (as I can see at least). If I understand correctly, it is Chr 17 and 23 that have iffy breakpoints not Chr 12?

>>>Two samples (CS5 and CS8) do not show any reads with the breakpoint pattern for Chr6 inversion. This could be because the reference sequence at the breakpoint contains gaps, indels, and misassembled region. We now explain this as a footnote in the new Table 2. In regard to Chr12, we carefully investigated the breakpoint region again and updated the table with our findings where all samples show the same breakpoint co-ordinates.

And most importantly the referring to Figure 1 in this regard is wrong – this does not show the near similar breakpoints. Think you should move the ref to Figure 1 to the lines above (119-120), since Figure 1 do show a schematic overview of the identification of the breakpoints. Here, I would say that the authors should really re-think how they present this overview. Figure 1 is a very complex figure and not easy to assess. I acknowledge that what you try to show is complex, but I think that with some edits this figure would be more understandable for the reader. For instance, instead of how it is now illustrated you could show how the PacBio read is split into two parts and how the orientation is shifted on the PacBio read itself and not on the reference (this is misleading I think). This can be shown below the actual reference with arrows (showing the PacBio read), and then from these arrows (showing the split) show the alignments onto the full PacBio read below here again ... think this could help on the readability of the figure

And the on top only have the candidate genes listed as before.

>>>We agree that Fig. 1 is complex to comprehend. We like the reviewer’s suggestion and have implemented it in the new version, which is now Fig. 2.

Further you state: “We re-evaluated the breakpoints obtained in our previous study for Chr6 and Chr17 inversions and found 1-3 kb shifts at two positions (Supplementary Table 2).” These shifts are not indicated in the table. Would suggest that you add a second column to show these shifts properly. And maybe state why you see these shifts. I would assume due to genome assembly differences (that the newer assemblies generated within this publication is of higher quality)?

>>>Reviewer is right that the shifts are due to better quality of the PacBio assemblies. We have now included this point in the new version (Lines 158-159) and also indicated it in Table 2.

Additionally, I would suggest that the confirmation of the N and S scaffolds generated by optical mapping data could be placed in the supplement (not needed as a main figure I think). Since quite similar to what we see in Figure 4.

>>>We agree with the suggestion and moved this figure to the supplement (Supplementary Fig. 3).

Then to the next section: “Identification of ancestral haplotypes using European sprat as an outgroup” which I suggest to move further down in the results section as a part of the “Origin of inversion haplotypes”.

These sections belong more or less together, I think. And I also want to point out that the authors should be a bit more cautious about their interpretations when only using an outgroup species as the main comparison to define ancestral vs derived state of an inversion. They cannot be 100% sure that this outgroup species does not harbour any inversions and/or SV in these genomic regions, and not at least which state of is represented in the genome sequenced.

That said, they do show that Pacific herring do not harbour the same inversion as Atlantic herring, which again support their hypothesis. However, the maximum likelihood trees of the inversion haplotypes and the IGV plots do indicate that also Pacific herring have some SV in these regions that should be looked into. And what is the state of this region (at the population level) in sprat? I would assume nobody has looked into yet? However, if the authors do have additional information here from the mentioned Pettersson M. E. et. al. manuscript that would be of importance for this interpretation – I would encourage the authors to give the reviewers get access to and/or insight into these results so that we can fully evaluate these statements.

>>>We have moved this paragraph so that it occurs just before “Origin of inversion haplotypes” and we think this is a logic order. We have also softened the language in this paragraph and changed “concluded” to “suggests” (Line 244). In sprat we have only long reads from one individual and pooled population sequencing based on short reads but these data give no indication for inversions in this region. The sprat paper has been submitted for publication and been uploaded to bioRxiv

(<https://www.biorxiv.org/content/10.1101/2024.02.16.580647v1.full>) so the information is available to reviewers. Furthermore, a high-quality, chromosome-level assembly of sprat has

recently been released by the Darwin Tree of Life project and we now take advantage of this to support our conclusions.

And some additional comparisons of the sprat genome vs Atlantic herring genome would have helped. Is the chromosome nr the same in these species? Is the homology/synteny OK between the homologous chromosomes? Based on this I would suggest that Figure 3 could have included additional dot plots of the all chromosomes and not only the inversion break points. This would also aid in the interpretation of the inversion region, and how solid these results are. If no further information is given, I would strongly suggest to tone down the statements in the ms, and define inversions as “putatively” ancestral and/or putatively derived. If Figure 3 should be part of the main ms or moved to the Extended data should be based upon if they are willing to show more of the comparative work here or not, I think.

>>> The reviewer is right that more information about synteny and homology between the two species would have been useful to determine the robustness of our results. A new reference genome for the European sprat has been released recently by Darwin Tree of Life and we decided to repeat our analyses to determine the ancestral state of each inversion with this new improved reference genome. We now include a synteny plot and dot plots for Chr6, Chr12, Chr17 and Chr23, showing synteny for the two reference genomes (Supplementary Figure 6). Approximately 30% of the 26 chromosomes of Atlantic herring map to the 20 chromosomes of the European sprat genome, with sequence identities in the range 0.25 and 0.75 (Ref 60. Pettersson et al. 2024, Supplementary Fig. 6b). The lack of perfect synteny between chromosomes could be due to relatively high divergence between the species (> 11 million years), but the European sprat remains the closest described outgroup species to Atlantic and Pacific herring (Wang et al 2022, Mol. Phylogenet. Evol.). Nevertheless, at least partial mappings between both species are found in the inversion regions, which are the basis for determining the ancestral and derived haplotypes (Fig. 4). Therefore, we used this new alignment to determine the ancestry and replaced original dotplot figure which had only breakpoint and smaller sequence with a new figure showing an entire inversion region (Fig. 4).

For the section “Structural variations at the breakpoint regions» that I suggest you to move up – I also have some specific comments:

In line 180-183 you write: “The sequence alignments of N and S alleles near the breakpoints indicated that the breakpoints for all four inversions were flanked by inverted duplications ranging from 8-60 kb in size and contained one or no gene (Fig. 4 and Extended datafig. 4).» As far as I can see from Figure 4 and other statements later on in the ms (see f. ex line 373-374) – this is not the case for the inversion on Chr 23. So, I would assume that you should correct this sentence!

And PS would also state that I truly love Figure 4! And think that moving this up would also be good for the ms itself (as part of moving this section up!). So, this now becomes Figure 2.

>>>We corrected the sentence from “four inversions” to “three of the four” and indicated that Chr23 breakpoint is difficult to interpret. We rearranged the figures as per suggestion.

Moreover, I really find your observation about the distal breakpoints being divergent between the individuals inspected as highly intriguing! You write in line 188-190: “For instance, distal breakpoints of all inversions were divergent among individuals, revealing the existence of non-shared structural variants. In particular, the Chr17 and Chr23 breakpoints were the most complex.”

However, these results are now only displayed by Figure 4, which is a comparison between two individuals, i.e. the CS10_hap1 and BS3_hap2 genome assemblies. Could this be plotted for the other individuals too? And shown as a Supp Figure?

>>>We have now cited Supplementary figures 4, 5 showing alignments of all inversion haplotypes with dotplots and genome graphs, respectively.

Additionally, I suggest that you also refer to the Supp Table 5 here (where you list all the different SV in more detail).

>>>Done!

And one additional question how different are these sequences? Would be good to have some estimations regarding the sequence homology.

>>>This is handled in the section “Origin of inversion haplotypes” and we report genetic distances (*da*) in Fig. 5, differentiation in Fig. 6 and *dxy* in Supplementary Fig. 9.

Further you state: “The distal breakpoint of Chr17 coincided with a telomeric sequence that varies in length (0-300 kb) outside the breakpoint and that is misaligned in the genome graph.”

This finding should have been visualized on the genome graph on Extended Figure 5 I think – do not catch this finding from the Figure as is. And I would also strongly suggest that for ALL figures that includes sequence/chromosome alignments to include bp position (such as Extended Figure 5 amongst others). On this figure the breakpoint position should also be noted.

>>> We have incorporated your suggestions in the genome graph figure (now Supplementary Fig. 5). We have also included bp positions in the figures whenever possible and for those figures where it was not possible, we mentioned the positions by other means such as arrowhead and in the figure legends.

For the next section where I have some issues is: “The evolutionary history of inversion haplotypes”. Here, in line 256-261 you state the following:

“The inversion regions showed strong differentiation between N and S homozygotes (high FST) which is in sharp contrast with the flanking regions (low FST). An exception to this is a region proximal to the Chr17 inversion breakpoint. A careful inspection of our PacBio data showed that this is not part of the inversion and must be a sequence polymorphism in very strong LD with the inversion polymorphism.” This careful inspection needs to be shown. What did you find? From the information given and what is displayed in Figure 6, 8 and Extended Data Figure 2 and 5 – this looks suspiciously like a miss-assembly – and that this should be part of the inversion: by flipping/reorient the region outside of the inversion. Especially when looking at the Extended Data Figure 2, both the IGV (first part of the inversion) and the sliding window PCA. Do you have PacBio reads spanning this region to confirm this or not. This needs to be revisited and/or better documentation for not being part of the inversion should be included!

>>>This high Fst region outside the inversion is not caused by a misassembly. The evidence for that is illustrated in the Supplementary Fig. 11 and in the figure attached to this comment. In Supplementary Fig. 11, we aligned the reference sequence to the sprat assembly from Darwin’s tree of Life, and to N and S alleles that are constructed using PacBio assembly and optical mapping data. From Han et. al., 2020, we know that reference has a Northern haplotype, which is why N allele shows linear alignment with the reference. From our ancestral state analysis in this paper, we know that N allele is ancestral for Chr17 inversion, which is why sprat sequence shows linear alignment with reference and S allele shows inverted alignment for the inversion sequence. Thus, we conclude that the high Fst region cannot be part of the inversion. Furthermore, the figure below (modified from Han et. al., 2020) shows that in the contrast between Atlantic and Baltic herring the genetic differentiation for the high-Fst region is much stronger than for the inversion, which is not expected if the high-Fst region was an integral part of the inversion.

Fig. Taken from Figure 3 – figure supplement 1 in Han et. al., 2020 comparing spring-spawning populations of Atlantic and Baltic herring.

For the IGV plot also here bp position should be denoted. Additionally, I would recommend that they display not only the inversion but spanning some parts outside of the inversions – by doing this it would be easier to see the inversions, and if the inversions are built up by more than one inversion (i.e. being a nested or double inversion). For me, this could in fact be the case for the inversion on Chr 17 and 23. I.e. the beginning of 23 as well as the beginning of 17 having a different pattern, indicating that they are built up by a double/nested inversion?

But here, I would for Chr17 look if this is due to a miss-assembly or not as I suggested above.

>>> Regarding the genotype plots of Extended Data Figure 2A (now Fig. 1), here we represent the genotype at diagnostic positions for Northern and Southern haplotypes ascertained from our previous study (Han et al 2020) that show high allele frequency differences between inversion haplotypes. We have tried to add the position numbers below the plot, but this results in a very crowded image, as the positions do not represent a continuous scale and cannot be summarized as an axis. The positions and code to reproduce the analysis are deposited in figshare and github for future reproducibility. Also, we tried to display positions outside the inversion, but as shown in Fig. 6, genetic differentiation outside the inversions is very low between haplotypes. We believe the F_{st} plots of Fig. 6 to be informative of this lack of variation. We made changes to the legend of Fig. 1 to be clear about what is represented in the heatmap.

PS: I truly like this figure (Extended Data Figure 2) – and would consider moving this one after some modifications to become part of the main ms.

>>>We agree, and we have modified the figure it is now included as Fig. 1.

Then for the inversions on Chr 17 and 23 showing a little less pronounced F_{st} and LD patterns than the inversion on Chr 6 and 12. Can you speculate why you see such differences between the inversions?

Can this difference be due to these ones being of an older age and/or younger age (even if this was not shown by the divergence estimations?). Or could it be due to these ones are built up of double inversions and increased recombination? Seems like you at least show that the linkage is weaker for these inversions.

>>>There are many possible explanations for these different patterns such as those mentioned by the reviewer. We have now added a comment that an explanation could be the distribution of polymorphisms under selection, if these occur primarily around the breakpoints we expect that genetic exchange (gene flux) may lead to a decay of F_{st} and LD patterns at the central part of the inversions as indicated for the Chr17 and 23 inversions (see Line 480-483)

In line 292-293 you write: “Furthermore, the observed derived alleles at high frequencies are

candidate mutations for being under positive selection.”

Not sure if this is part of your observations/results or as an intro to the following sentence: “Further, we compared the dN/dS ratio for the N and S alleles at each locus in an attempt to find genes that may show accelerated protein evolution as part of the evolution of these adaptive haplotypes.”

Please revisit this part of the text and clarify.

>>>We have deleted the sentence ”Furthermore,---”. It must be a left-over after a revision we did before submitting the paper. Thanks for spotting this.

If Figure 7 is needed as part of the main ms can be re-evaluated, I think. Could easily be moved to the Extended Data Section.

>>>We prefer to keep it as a main figure because we think this is an important finding in this paper. The literature on supergenes/inversions always discusses the accumulation of load and these are nice illustrations that we do not see any signs of this for the herring inversions

For the intro of the discussion: I would say that the authors should try to shorten it a bit but also here in the intro mention that they have done a detailed characterization of the breakpoint regions and thus, enable to define how the originated (by ectopic recombination). This is as I see it an important part of the work conducted and now missing from the intro of the Discussion.

>>>We have shortened the intro of the discussion and also included the role of ectopic recombination.

Then, I have one additional question. Did not the optical mapping conducted on CS10 and BS3 generate chromosome level genome assemblies? You do display results of the whole chromosomes of Chr 6, 12, 17 and 23. I guess you also have the whole chromosomes for the other chromosomes? If yes, the fact that you do generate two new high quality reference genomes for Atlantic herring should be highlighted.

This will be a really nice resource that should be released for the research community, I think!

In this regard I also want to point out that in line 530-531 you write:

“However, the resulting hybrid scaffolds had many gaps and were not contiguous for the entire inversion regions (Supplementary Table 1).”

The hybrid genomes are not listed in this table ... Please provide the data needed to show this.

>>>Genome statistics for assemblies generated using optical mapping are in Supplementary Table 10. We apologize for citing the wrong table. Based on these statistics and manual inspection of certain regions, we think that these assemblies are not of sufficient good quality to be used by the research community. The assemblies contain many gaps and unscaffolded sequence. We manually inspected all inversion regions and closed gaps in order to be precise in our analysis. (By the way we are preparing a new reference assembly based on deep PacBio HiFi sequencing that gives very high contiguity that will be released later this year).

PS: This is also the case for other results presented I think – that the authors in several occasions refer to tables and figures that do not show/display the results that they refer to. And maybe more importantly several times also describe results that have no table or figure at all. Please go through the entire ms – and present the result that is needed in a proper way.

>>>We have followed this advice and carefully checked the new version of the manuscript.

Finally, and maybe most importantly, the authors need to revisit their definitions of the breakpoints, I think. First, of all not sure why you have defined that the inversions should not harbour the inverted duplications in the distal end for Chr 6, 12, and 17 and for some of the inversions also in the proximal end (for Chr 6 and 12)?

For me this seems rather odd since these genomic regions are inverted and should in my opinion part of the inversion. This is also how it seemingly looks like when looking at Figure 4. If this is not what you see when using the PacBio reads this should be presented in the ms as a figure/table, I think.

In this regard I also want to mention: When looking at some of the figures the definition of the

breakpoints seems not to be consistent. I suggest that you have a closer look at this once more. F. ex for Chr 17 on Figure 6 the light grey marking show that the inversion spans the entire distal end of that Chromosome. This is not as is shown on Figure 4, Figure 2 nor the Extended Data Figure 4. But what you see when you look at the LD in Figure 6 as well as what you see on Figure 8 (the dAF' analyses) and the Supp Figure 2. Please revisit this. And for the Supp Figure 2 I also notice that you here have included the suspicious region (that I mentioned above) as part of the inversion, which I think is correct. But if this is the case, this needs to be looked into and changed in other part of the ms.

>>> Firstly, we define the inversion breakpoints at the position where the orientation of the inversion sequence switches as described on line 142-145. Thus, the sequence of the WT and Inversion alleles are colinear outside our definition of the inversion breakpoints. We took a closer look again as suggested and analyzed the inverted duplicate sequences from *N* and *S* alleles and included that comparison in Supplementary Table 5.

Secondly, the reviewer highlights inconsistencies between figures when representing inversion breakpoints which we have thoroughly revised and corrected. For the specific case of Chr17 and the discrepancy between Figure 4 (now Fig. 3), Figure 2 (now Supplementary Fig. 4) and Extended Data Figure 4 (now Supplementary Fig. 5) vs Figure 6, we would like to highlight that the first three figures are based on the alignments of the PacBio assemblies whereas Figure 6 is based on the alignment of short read data to the Atlantic herring reference genome. We describe in lines 210-212 that the PacBio assemblies recover 0 to 300 kb more of the telomere region of Chr17 compared to the reference genome, which explains why figures based on PacBio assemblies sometimes extend beyond the distal breakpoint of the reference. With figures based on short read data mapped to the reference genome, statistics are only calculated until the distal breakpoint of Chr17. We have tried to make this clearer legend of Figures 3, 6, and 8.

PS. For Figure 6 C and D the Chromosome naming is lost (not shown), probably hidden by Figure 6E?

>>>This has been corrected.

And as a last comment: I again have to point out that I truly enjoyed reading the ms and I really think that the work presented is substantially improving our knowledge on the evolutionary role of larger chromosomal inversions.

>>>Thanks, we appreciate the comment very much.

Reviewer #2 (Remarks to the Author):

This manuscript presents the long-read assemblies of 12 Atlantic herring individuals and an analysis of population short-read re-sequencing analysis of 49 Atlantic herring and 30 Pacific herring. This study extends the work of Han et al (2020) that identified four major inversions that differentiate northern versus southern populations of Atlantic herring. Han et al (2020) did a short-read sequencing study of 35 to 100 herring and used the data to identify population specific genetic differences. The inversions were identified based on clusters of differentiated SNPs between the populations. Han et al. (2020) suggested that major shifts in allele frequencies at candidate loci within the inversions bear molecular population genetic signatures of selection that are correlated with environmental differences between northern and southern populations such as temperature and salinity.

This study uses long-read read sequencing to assemble genomes that identify inversion breakpoints in the northern (Baltic Sea n=6) and southern (Celtic n=6) Atlantic herring collections. The goals of this study are:

1. to determine the mechanisms that generate inversions by mapping breakpoint coordinates and structural variants (SVs) around the breakpoints;
2. to infer the origin of inversions by determining its ancestral state and age using European sprat

(*Sprattus sprattus*) as an outgroup species;

3. to use phylogenetic analysis to infer the evolutionary history of inversions;
4. to examine the effects of suppressed recombination by analyzing patterns of variation, differentiation, linkage disequilibrium, mutation load, and gene flux (genetic exchange between inversion haplotypes).

The major findings of the paper are:

1. The paper describes the generation of 12 high quality long-read assemblies of Atlantic herring. One of the challenges for assembly construction were for individuals heterozygous for the northern and southern inversions. The assembly algorithm hifiasm did create primary and secondary contigs for the two arrangements, however, HiCanu did not infer contiguous sequence through the breakpoints of the secondary arrangements. Optical genomic maps were used to confirm the assemblies and scaffold contigs across the inverted regions.
2. Breakpoints were mapped for four inversion events on four chromosomes. The derived inversion is of unique origin based on shared breakpoint coordinates for three of the four inversion events. Inverted repeats are found within the breakpoint regions suggesting that pairing of the repeats and ectopic exchange generated the inversions. None of the breakpoints disrupted the coding regions of genes rejecting one aspect of the unreferenced position effect hypothesis for the establishment of inversions (KIRKPATRICK AND BARTON 2006). The position effect hypothesis cannot be completely ruled out because this study does not present transcriptomic data to test whether the breakpoints disrupted gene expression of boundary genes (PUIG et al. 2004).

>>> Thank you for pointing out this important conclusion of our study. In fact, as we do not find any disruption of coding sequences at the breakpoints (genes and their coordinates surrounding the inversion breakpoint are reported in Supplementary Table 3 and are also used to place genes on the reference chromosome in Fig. 2), the functional impact of the inversion is therefore expected to be altered gene regulation. We have added this conclusion to lines 528-529.

3. The ancestral and derived arrangements was inferred for the four inversion polymorphisms using the sprat outgroup species. The four inversion polymorphisms are > 1.5 million years old, which is consistent with the comparable levels of nucleotide diversity between ancestral and derived arrangements. Ancestral and derived arrangements are highly differentiated from each other. Consistent with this is the high levels of linkage disequilibrium, but not all sites are in LD with each other, which suggests that gene flux may be breaking up LD as has been seen in *Drosophila* (FULLER et al. 2017).

>>>We discuss the evidence of gene flux on Line 468 and onwards.

4. Divergent selection was inferred based on maintenance of alternative haplotypes in the face of high gene flow, however, no data or citations are provided to support the contention that gene flow is sufficient to prevent differentiation.

>>>We have now estimated gene flow, see response to your Comment 10.

5. No evidence for genetic load based on dn/ds ratios, Site frequency spectra are similar for synonymous and nonsynonymous sites and the proportion of TEs between N and S inversions is similar.
6. There is evidence for genetic flux among inversion types based on allele sharing among arrangements. There is mixture of sites that show lack of exchange with sites that show extensive sharing. Sharing is uniform across the inverted regions of the four inversion polymorphisms. This pattern is consistent with the LD plots that show low and high LD among sites. This pattern is expected for old inversion polymorphisms.
7. The inversion polymorphisms shows clinal variation associated with different environments.

Strengths. This manuscript presents new long-read sequence assemblies for Atlantic herring that have

four inversion polymorphisms on four chromosomes. Studies of inversions until recently have been limited to species with methods that allow detection with direct approaches. Comparative genomics with techniques like long read sequence allow the detection of inversions in non-model organisms. The data presented here allow the analysis of inversion breakpoints between ancestral and derived arrangements providing insights into the mutational mechanism that generates inversions. These data also allow the inference of the ancestral and derived arrangements as well as estimates of the time of origin of the chromosome. Population re-sequencing data provide data on the structure of nucleotide diversity within and between the arrangements.

Weaknesses. The paper could do better job of framing the work as testing hypotheses about the origin, establishment, and maintenance of inversions (KIRKPATRICK AND BARTON 2006). Figure 1 that shows the comparison of ancestral and derived inversions is confusing. The paper would benefit from analysis of gene flow to make the case for haplotype differentiation in the face of gene flow.

>>>We now present an improved Fig. 1 (Fig. 2 in the revised version) and we include an analysis of gene flow. Also, we discuss the hypothesis about the origin, establishment and maintenance of inversions in more detail (see Line 522 and onwards)

Overall assessment: This paper addresses important questions in evolutionary biology, i.e., how do karyotypic variants arise in populations and how are they established and maintained. Until genomic approaches were developed, the number of systems where chromosomal evolution could be studied were limited. With the ability to sequence complete genomes now allows chromosomal evolution to be studied in diverse organisms such as the Atlantic herring in this study. The overall analyses are sound and are reproducible based on the described methods and the conclusions follow from the analyses.

Comments

1. Lines 36-37. Whether alleles within an inversion are inherited as a single unit needs to be tested. There is evidence from *Drosophila* that homologous gene conversion can transfer alleles between different inversion backgrounds despite reduced levels of crossing over.

>>>We think the sentence is OK because we say that sets of alleles are inherited as a single unit, we do not say that all sets of alleles within the inversion are inherited as a single unit. A key feature of inversions is high LD among sets of alleles within the inversion. Furthermore, on Line 59-61 we acknowledge genetic exchange between inversion haplotypes. The data presented in this paper reveal a dynamic evolution of inversion haplotypes as previously documented in *Drosophila*. No change.

2. Lines 139-140 Figure 1. This figure is confusing. For example in A, the chromosome 6 inversion is 2.5 Mb long, yet a single 25 kb PacBio read is shown mapped to breakpoints on both ends of the inversion. I assume that the reference genome is one inversion type and the PacBio Read comes from the alternative inversion type, but this is not clear from the Figure legend. An additional issue is why is only one breakpoint PacBio read shown for each inversion event. It would also help to show the inferred coding genes on the PacBio read. Calvete et al (2012) showed that gene duplications and deletions of genes can accompany inversion events in *Drosophila*. This may not necessarily be the case, but the paper should indicate whether gene structure at the breakpoint boundaries is conserved or not. The dot plots in the boxes in B and C are hard to see.

>>>We have simplified Fig. 1 (now Fig. 2). It is true that we analyzed PacBio reads with the alternate inversion haplotype that of the reference. We have now indicated this in the figure legend. We also changed the dotplots where the line is darker.

3. Lines 160-172 and 577-584. Inferring the ancestral state of the inversions. This section could be shortened by skipping the discussion of using levels of nucleotide diversity to infer ancestral versus derived state. The nucleotide diversity analysis may have been done to infer the derived arrangement, but because it did not provide a definitive answer, the text from Line 161 to 167 can be deleted. A more direct method is to use the configuration of genes adjacent to the breakpoints as markers of the inversion event using *Sprattus sprattus* to polarize the mutation event, which was done here. If

Sprattus sprattus has a/b and c/d at the proximal and distal breakpoints, then the derived inversion will likely have the configuration a/c and b/d at the proximal and distal breakpoints (See Supplemental Figure S4 showing the conversion of the Standard to Arrowhead arrangement for an example. FULLER et al. 2017).

>>>We have shortened this text to a single sentence (Line 233-235).

4. Line 198 and Line 390. Are the repeats at the breakpoints of the different inversions similar to each other suggesting a common mutational motif?

>>>We have now analyzed these repeat sequences in more detail and found that these sequences are not similar among different inversions. We have now mentioned this observation on Line 201-202.

5. Line 261-262 and Figure 6. Was linkage disequilibrium tested between all pairs of SNPs?

>>> To be able to generate the R^2 heatmaps from vcfTools output, we thinned the SNPs within the region by keeping only SNPs with genotype quality above 30, less than 10% missing data across all individuals, $maf > 0.1$ and that distanced at least 5 kb from each other. We now explain this better in the Materials and Methods (Lines 787-789) and make a link to M&M in the legend of Fig. 6.

6. Lines 310-312. Are the site frequency data presented for just the genes within the inverted regions? If so, it would be useful to contrast these data with collinear regions. If the SFS are similar, then this would support the conclusion that inverted regions are not accumulating deleterious alleles, i.e., no genetic load.

>>>The genome-wide data are presented in Supplementary Fig. 12) as explained in the main text: "Further, the site frequency spectrum of *N* and *S* homozygotes were similar to each other (Fig. 7b) and to that of the genome-wide estimate (Supplementary Fig. 12)" (Line 337-339 in the revised version. We clarify that only genes inside the inversions are used in Fig. 7 in the legend.

7. Lines 281-297 It would be useful to know what the frequencies of the three inversion genotypes are for each inversion polymorphism in the different regions. Are they in HWE? If there are deleterious alleles present, one might expect to observe a deficiency of homozygotes and overabundance of heterozygotes.

>>> Thank you for this suggestion. We have included calculation of Hardy-Weinberg equilibrium for all inversions and populations in this study and find no significant deviation (albeit our sample sizes per population are low). This suggests that we do not find an excess of heterozygotes; on the contrary, in many populations we observe the fixation of one of the inversion alleles (see Supplementary Table 9 and Figure 4 - supplementary figure 3 from Han et al 2020). This calculation is presented in Supplementary Table 9 and mentioned in the discussion (Lines 524-525).

8. Lines 403-408. The U-shaped pattern of divergence is expected for neutral inversion polymorphisms (NAVARRO et al. 2000). A non-U-shaped pattern might reflect selected genes within the central part of the inverted region that are responsible for the establishment and maintenance of the different arrangements (GUERRERO et al. 2012). In addition to the *D. subobscura* example, consider Figure 7 in Fuller et al. (2017) as example of the maintenance of differentiation in central inverted regions.

>>>We agree and we have revised this text on Line 477-496 and we added the citation to Fuller et al. (2017) on Line 483.

9. Lines 414-419. The maintenance of differentiation in the face of extensive gene flux supports the idea that selection is removing transferred genes that create maladaptive phenotypes.

>>>We completely agree and has clarified this in the revised version (Lines 493-496).

10. Lines 439-443. What is the evidence for gene flow? The data presented here could be used to

estimate the migration parameter Nm , where N is the effective population size and m is the migration rate. One would want to use data from non-inverted chromosomes to estimate Nm . If Nm is greater than 1, than migration is sufficient to homogenize gene frequencies genome-wide. Genes or inversions that are differentiated in the face of this level of gene flow are likely to be selected (HOEKSTRA et al. 2005).

>>> Thank you for the suggestion to include an estimation of gene flow in our manuscript. We agree that such an analysis strengthens our argument. We now include a pairwise estimation of F_{st} and Nm for all populations based on genome-wide data excluding inversion regions, based on Slatkin 1993. The levels of estimated population migration rates are well above 1 (and F_{st} values very low), suggesting that gene flow is strong enough to homogenize differentiation between herring populations. In addition, we now also include Hardy-Weinberg calculations for all inversions and populations and find no significant deviations (Supplementary Table 9) and thus, no excess of heterozygotes indicative of overdominance as is noted for some supergenes. High F_{st} in the face of gene flow and no deviation from HWE (no excess of heterozygotes) suggest that the inversions are maintained by divergent selection. We now include this argument in the discussion (Lines 512-518)

11. Lines 730-744. Betran et al. (1997) developed a method to infer genetic flux in the form of gene conversion and is implemented in DNASP (ROZAS et al. 2017). The method infers conversion tracts using similar reasoning as described here, but allows one to estimate the gene conversion parameter.

>>>We think the use of this method is out of the scope of this paper.

Editorial Comments

1. Line 42. Change “single unit” to “single copy” The paper is referring to the unique occurrence of a new inversion mutation where the chromosome breaks at two sites and is rejoined in reverse orientation.

>>>Done

2. Lines 114-115 Table 1 and Supplemental Table 1A. There should be a reference to BUSCO for the completeness analysis (SIMÃO et al. 2015)

>>>The reference is cited in the Methods section but we have added the reference to this table as well.

3. Line 372. A citation to Richards et al. (2005) could be added here for another example from *Drosophila*

>>>We have added this reference, which is now on Line 423.

4. Lines 1013-1014 Table 1 is duplicated.

>>>This has been fixed

REVIEWERS' COMMENTS

Reviewer #1 (Remarks to the Author):

Dear Jamsandekar et al.,

I have now had the pleasure to read the revised version of the ms entitled "The origin and maintenance of supergenes contributing to ecological adaptation in Atlantic herring". I have to say that I really appreciate the effort made to further improve the ms in relation to both to mine and the second reviewer. You have done a respectful and thorough job here! The ms have improved a lot! However, I still have some minor issues that I want you to take a closer look at.

First of all, Figure 2 (previously Figure 1) is still not of satisfying quality, I think. It is still rather confusing, and modifications are needed before any resubmission. I have made some suggestions how you could visualize your results in a better way (only done if for Chr 6 ... but then show you how this could be done accordingly for all the other inversion chromosomes too). Easier for me to do it like this than try to do it by writing.

NB 1: Mine is still of rather low quality. Would encourage the authors to strive to make this figure of higher quality, since it is planned to be published in Nature Comm!

NB 2: For Chr 6 and 17 the visualization of the collinear region (outside of the inversion) as reverted, while the inverted part as forward is misleading. For these two figures (a and c), the PacBio read should have been reverse complemented before the alignment is done. As visualized in my plot for Chr 6. This is not needed for Chr 12 and Chr 23.

And by doing this reverse complement, it is rather obvious that the unlaced scaffold (unplaced_scaffold (un_sc) 699) should be placed as part of the collinear sequence in front of the breakpoint region (see my suggestion for Figure 2a here). This should need be taken into consideration in the revised ms (and PS this will then shift the start of the breakpoint region).

Furthermore, in my suggestion (as you can see) I also added a schematic overview of the alternate variant (for the reader to more easily understand the orientation and alignment of the single PacBio read) in addition to the orientation of the genome assembly variant. And most importantly, the orientation of the colinear vs. the inversion variant. So, would really encourage you do revise how to present Figure 2, based on my comments (and please feel free to use my suggestions as inspiration if you like).

Then, over to some other minor issues that should be corrected/looked into before any re-submission. The line nr listed below, are referring to the line nr found in the resubmitted version (without track changes):

On line 126-127 you state:

"Each line represents one of 35 individuals from the Baltic and Celtic Sea (dark green and yellow individuals from panel a.)"

Here an end parenthesis is lacking:

“Each line represents one of 35 individuals from the Baltic and Celtic Sea (dark green and yellow individuals from panel a).”

On line 137-138 you write:

“Characterization of inversion breakpoints using PacBio read alignments on chromosomes 6, 12, 17, and 23”

Here, you have added “using PacBio read alignments” in the title, not sure if this addition is needed here? I would suggest to remove, since this section also cover the use of the PacBio assemblies.

Additionally, I also see that you here use PacBio read alignments while later in the section use “HiFi” read only .

Think you should use should strive to use both HiFi PacBio since this is the most precise wording (throughout the ms), and most importantly in this section where this has only briefly been mentioned in the intro before this.

In connection to this, I think line 139-146:

“We used the PacBio read alignments to the reference assembly as well as PacBio genome assemblies to study inversion breakpoints, where the former helped us to accurately define breakpoint coordinates and the latter helped us to discern the sequences at the breakpoints leading to the formation of inversions. In case of read alignments to the reference assembly, reads representing the alternate inversion allele spanning the inversion breakpoint is expected to show a particular pattern where the reads get split into two parts, one part aligns outside the inversion and the other part aligns inside the inversion at the opposite end in reverse orientation. The alignment of such HiFi reads is presented in Fig. 2.”

would have benefitted from some clarifications/modifications (see below where I have made some suggestions):

“The PacBio HiFi read alignments (towards the reference assembly) in combinations with the PacBio genome assemblies were used to investigate the inversion breakpoints in detail. First, we used single read alignments to accurately define inversion breakpoint coordinates. Here, the reads representing the alternate inversion is expected to show a particular pattern where the reads get split into two parts: one aligning outside the inversion and the other aligning inside the inversion at the opposite end in reverse orientation. Alignments of such PacBio HiFi reads are presented for each of the chromosomes harboring an inversion (see Fig. 2).” Secondly, we used the PacBio genome assemblies to discern the sequences at the breakpoints leading to the formation of inversions.

On line 153-155 you write:

“Chr6 slightly deviated from this common observation as we found the distal breakpoint for one of the samples to be 500 kb further along the chromosome (Supplementary Fig. 2, Table 2).”

This could be modified/specified:

“However, it should be noted that for the inversion on Chr6, the distal breakpoint slightly deviated from this common observation for one of the samples, with being 500 kb localized

further upstream on the chromosome (Supplementary Fig. 2, Table 2).”

For line 179-181 you state:

“A part of the HiFi read maps to unplaced_scaffold (un_sc) 699 (indicated by yellow dotted lines) as the reference assembly is not scaffolded at this region.”

Please have a look at the placement of this scaffold, if it should be placed in front of the breakpoint region (as I have suggested in my figure). And if so please re-write.

For the footnote of Table 2 you write:

“CS5 and CS8 samples had no reads spanning the breakpoints of chromosome 6 inversion because the reference sequence at the breakpoint had gaps, SVs, and misassembled sequence.”

Maybe rewrite a bit?

“For the samples with ID CS5 and CS8 the co-ordinates for the inversion breakpoints on Chr 6 is not given, due to the lack of reads spanning these regions within the respective reference assemblies.”

On line 190 you have given this title:

“Structural variations at the breakpoint regions and origin of inversions”

Here I suggest to drop the new add on: “and origin of inversions” mainly since this is not touched upon in this result section, even if you use these results to touch upon this in the discussion. Which is perfectly fine.

On line 200-201 you write:

“However, such highly identical sequence was not found on the S allele (Supplementary Table 5).”

I suggest to specify that there is more than one S allele:

“However, such highly identical sequences were not found on the S alleles (Supplementary Table 5).”

On line 232-236 you write:

“The estimates of nucleotide diversity (π) for the N and S haplotypes are similar for all four inversions³¹, which means that these estimates are not informative about which allele is likely to be derived. We therefore decided to use the European sprat (*Sprattus sprattus*) as an appropriate outgroup species^{38,39} to determine which inversion haplotype represents the derived state because a high-quality reference genome has recently been released⁴⁰.”

Maybe rewrite a bit?

“Previous reports have shown that estimates of nucleotide diversity (π) for the N and S haplotypes are similar for all four inversions³¹, and thus, cannot be used to identify the ancestral vs. the derived state of the inversion haplotype. To overcome this obstacle, we here use the recently released high-quality reference genome of European sprat (*Sprattus sprattus*)⁴⁰ as an outgroup species^{38,39}, and by such determine which of the inversion

haplotypes represent the ancestral vs. derived state.”

PS: I also suggest that you go over the rest of the text in this section once more.

On line 253 you have the title:

“Origin of inversion haplotypes”

I suggest to edit to include “timing/dating”:

“Timing of the origin of inversion haplotypes”

On line 623:

“In our previous study, we used PacBio Continuous long reads data from one ...”

I suggest to edit to:

“In our previous study, we used PacBio continuous long reads data from one ...”

Finally, I want to emphasize that I truly like the revised version of this ms, you have done a great job! But I would suggest that you read through the ms carefully once more, and have a look at all my comments/suggestions, that are meant to aid in further improving the ms.

Chr 6

Response to the Reviwer's comments

First, we would like to sincerely thank the reviewer for his/her efforts to help us to improve the paper, and for the positive comments. Our responses to the specific comments are given below.

I have now had the pleasure to read the revised version of the ms entitled "The origin and maintenance of supergenes contributing to ecological adaptation in Atlantic herring". I have to say that I really appreciate the effort made to further improve the ms in relation to both to mine and the second reviewer. You have done a respectful and thorough job here!
The ms have improved a lot! However, I still have some minor issues that I want you to take a closer look at.

First of all, Figure 2 (previously Figure 1) is still not of satisfying quality, I think. It is still rather confusing, and modifications are needed before any resubmission. I have made some suggestions how you could visualize your results in a better way (only done if for Chr 6 ... but then show you how this could be done accordingly for all the other inversion chromosomes too). Easier for me to do it like this than try to do it by writing.

>>>

NB 1: Mine is till of rather low quality. Would encourage the authors to strive to make this figure of higher quality, since it is planned to be published in Nature Comm!

>>>**We upload high-quality images with the final version**

NB 2: For Chr 6 and 17 the visualization of the collinear region (outside of the inversion) as reverted, while the inverted part as forward is misleading. For these two figures (a and c), the PacBio read should have been reverse complemented before the alignment is done. As visualized in my plot for Chr 6. This is not needed for Chr 12 and Chr 23.

And by doing this reverse complement, it is rather obvious that the unlaced scaffold (unplaced_scaffold (un_sc) 699) should be placed as part of the collinear sequence in front of the breakpoint region (see my suggestion for Figure 2a here). This should need be taken into consideration in the revised ms (and PS this will then shift the start of the breakpoint region).

Furthermore, in my suggestion (as you can see) I also added a schematic overview of the alternate variant (for the reader to more easily understand the orientation and alignment of the single PacBio read) in addition to the orientation of the genome assembly variant. And most importantly, the orientation of the colinear vs. the inversion variant.

So, would really encourage you do revise how to present Figure 2, based on my comments (and please feel free to use my suggestions as inspiration if you like).

>>>We have followed your advice and used your draft as an inspiration. We agree this is a much better illustration, thank you so much for this constructive criticism!

Then, over to some other minor issues that should be corrected/looked into before any re-submission. The line nr listed below, are referring to the line nr found in the resubmitted version (without track changes):

On line 126-127 you state:

"Each line represents one of 35 individuals from the Baltic and Celtic Sea (dark green and yellow individuals from panel a.)"

Here an end parenthesis is lacking:

“Each line represents one of 35 individuals from the Baltic and Celtic Sea (dark green and yellow individuals from panel a).”

>>>**We have corrected this mistake.**

On line 137-138 you write:

“Characterization of inversion breakpoints using PacBio read alignments on chromosomes 6, 12, 17, and 23”

Here, you have added “using PacBio read alignments” in the title, not sure if this addition is needed here? I would suggest to remove, since this section also cover the use of the PacBio assemblies.

>>>**We have shortened the title as suggested**

Additionally, I also see that you here use PacBio read alignments while later in the section use “HiFi” read only .

Think you should use should strive to use both HiFi PacBio since this is the most precise wording (throughout the ms), and most importantly in this section where this has only briefly been mentioned in the intro before this.

In connection to this, I think line 139-146:

“We used the PacBio read alignments to the reference assembly as well as PacBio genome assemblies to study inversion breakpoints, where the former helped us to accurately define breakpoint coordinates and the latter helped us to discern the sequences at the breakpoints leading to the formation of inversions. In case of read alignments to the reference assembly, reads representing the alternate inversion allele spanning the inversion breakpoint is expected to show a particular pattern where the reads get split into two parts, one part aligns outside the inversion and the other part aligns inside the inversion at the opposite end in reverse orientation. The alignment of such HiFi reads is presented in Fig. 2.”

would have benefitted from some clarifications/modifications (see below where I have made some suggestions):

“The PacBio HiFi read alignments (towards the reference assembly) in combinations with the PacBio genome assemblies were used to investigate the inversion breakpoints in detail. First, we used single read alignments to accurately define inversion breakpoint coordinates. Here, the reads representing the alternate inversion is expected to show a particular pattern where the reads get split into two parts: one aligning outside the inversion and the other aligning inside the inversion at the opposite end in reverse orientation. Alignments of such PacBio HiFi reads are presented for each of the chromosomes harboring an inversion (see Fig. 2).” Secondly, we used the PacBio genome assemblies to discern the sequences at the breakpoints leading to the formation of inversions.

>>>**We like this rewording and have made this change**

On line 153-155 you write:

“Chr6 slightly deviated from this common observation as we found the distal breakpoint for one of the samples to be 500 kb further along the chromosome (Supplementary Fig. 2, Table 2).”

This could be modified/specified:

“However, it should be noted that for the inversion on Chr6, the distal breakpoint slightly deviated from this common observation for one of the samples, with being 500 kb localized further upstream on the chromosome (Supplementary Fig. 2, Table 2).”

>>>**We have followed this advice and changed the text as follows:**

However, it should be noted that for the inversion on Chr6, the distal breakpoint slightly deviated from this common observation for one of the samples, with being 500 kb further along the chromosome (Supplementary Fig. 2, Table 2).

For line 179-181 you state:

“A part of the HiFi read maps to unplaced_scaffold (un_sc) 699 (indicated by yellow dotted lines) as the reference assembly is not scaffolded at this region.”

Please have a look at the placement of this scaffold, if it should be placed in front of the breakpoint region (as I have suggested in my figure). And if so please re-write.

>>>You are right and this has been corrected in the revised Fig. 2. We have moved this comment to the figure legend.

For the footnote of Table 2 you write:

“CS5 and CS8 samples had no reads spanning the breakpoints of chromosome 6 inversion because the reference sequence at the breakpoint had gaps, SVs, and misassembled sequence.”

Maybe rewrite a bit?

“For the samples with ID CS5 and CS8 the co-ordinates for the inversion breakpoints on Chr 6 is not given, due to the lack of reads spanning these regions within the respective reference assemblies.”

>>>We have modified the footnote as follows (change underscored):

The samples with ID CS5 and CS8 had no aligned reads spanning the breakpoints of chromosome 6 inversion because the reference sequence at the breakpoint had gaps, SVs, and misassembled sequence.

On line 190 you have given this title:

“Structural variations at the breakpoint regions and origin of inversions”

Here I suggest to drop the new add on: “and origin of inversions” mainly since this is not touched upon in this result section, even if you use these results to touch upon this in the discussion. Which is perfectly fine.

>>>OK, we have followed this recommendation

On line 200-201 you write:

“However, such highly identical sequence was not found on the S allele (Supplementary Table 5).”

I suggest to specify that there is more than one S allele:

“However, such highly identical sequences were not found on the S alleles (Supplementary Table 5).”

>>>Agreed

On line 232-236 you write:

“The estimates of nucleotide diversity (π) for the N and S haplotypes are similar for all four inversions³¹, which means that these estimates are not informative about which allele is likely to be derived. We therefore decided to use the European sprat (*Sprattus sprattus*) as an appropriate outgroup species^{38,39} to determine which inversion haplotype represents the derived state because a high-quality reference genome has recently been released⁴⁰.”

Maybe rewrite a bit?

“Previous reports have shown that estimates of nucleotide diversity (π) for the N and S haplotypes are similar for all four inversions³¹, and thus, cannot be used to identify the ancestral vs. the derived state

of the inversion haplotype. To overcome this obstacle, we here use the recently released high-quality reference genome of European sprat (*Sprattus sprattus*)⁴⁰ as an outgroup species^{38,39}, and by such determine which of the inversion haplotypes represent the ancestral vs. derived state.”

PS: I also suggest that you go over the rest of the text in this section once more.

>>>**Agreed, we have changed the text accordingly.**

On line 253 you have the title:

“Origin of inversion haplotypes”

I suggest to edit to include “timing/dating”:

“Timing of the origin of inversion haplotypes”

>>>**Agreed**

On line 623:

“In our previous study, we used PacBio Continuous long reads data from one ...”

I suggest to edit to:

“In our previous study, we used PacBio continuous long reads data from one ...”

>>>**We have made this change**

Finally, I want to emphasize that I truly like the revised version of this ms, you have done a great job! But I would suggest that you read through the ms carefully once more, and have a look at all my comments/suggestions, that are meant to aid in further improving the ms.

>>>**Thanks**